# Multisensory flicker modulates widespread brain networks and reduces interictal epileptiform discharges

Lou T. Blanpain [1,2,3], Eric R. Cole[1,3], Emily Chen[1], James K. Park[1], Michael Y. Walelign[4], Robert E. Gross[1,5], Brian T. Cabaniss[6], Jon T. Willie [7] ✉ & Annabelle C. Singer [2,3] ✉

Modulating brain oscillations has strong therapeutic potential. Interventions that both non-invasively modulate deep brain structures and are practical for chronic daily home use are desirable for a variety of therapeutic applications. Repetitive audio-visual stimulation, or sensory flicker, is an accessible approach that modulates hippocampus in mice, but its effects in humans are poorly defined. We therefore quantified the neurophysiological effects of flicker with high spatiotemporal resolution in patients with focal epilepsy who underwent intracranial seizure monitoring. In this interventional trial (NCT04188834) with a cross-over design, subjects underwent different frequencies of flicker stimulation in the same recording session with the effect of sensory flicker exposure on local field potential (LFP) power and interictal epileptiform discharges (IEDs) as primary and secondary outcomes, respectively. Flicker focally modulated local field potentials in expected canonical sensory cortices but also in the medial temporal lobe and prefrontal cortex, likely via resonance of stimulated long-range circuits. Moreover, flicker decreased interictal epileptiform discharges, a pathological biomarker of epilepsy and degenerative diseases, most strongly in regions where potentials were flicker-modulated, especially the visual cortex and medial temporal lobe. This trial met the scientific goal and is now closed. Our findings reveal how multi-sensory stimulation may modulate cortical structures to mitigate pathological activity in humans.

While focal deep brain stimulation is increasingly applied to treat neurological conditions that impact widespread brain networks such as epilepsy[1,2] and Alzheimer's disease (AD)[3], a possibly more optimal neurostimulation approach would be less invasive while still modulating broad brain networks. The brain's natural tendency to respond to dynamic sensory stimuli may be leveraged toward therapeutic applications. Rhythmic neural activity, or brain oscillations, play key roles in many brain functions including sensory processing,

[1]Department of Neurosurgery, Emory University School of Medicine, Atlanta, GA, USA. [2]Neuroscience Graduate Program, Graduate Division of Biological and Biomedical Sciences, Emory University, Atlanta, GA, USA. [3]Coulter Department of Biomedical Engineering, Georgia Institute of Technology & Emory University, Atlanta, GA, USA. [4]Department of Electrical and Computer Engineering, Georgia Institute of Technology, Atlanta, GA, USA. [5]Departments of Neurosurgery and Neuroscience and Cell Biology, Rutgers Robert Wood Johnson Medical School, New Brunswick and New Jersey Medical School, Newark, NJ, USA. [6]Department of Neurology, Emory University School of Medicine, Atlanta, GA, USA. [7]Departments of Neurological Surgery, Neurology, Psychiatry, and Biomedical Engineering, Washington University, St. Louis, MO, USA. ✉e-mail: jontwillie@wustl.edu; asinger@gatech.edu

attention, and memory[4–11], and are impaired in neuropsychiatric disorders as diverse as epilepsy, AD, schizophrenia, and autism[12]. Promoting or restoring specific oscillations associated with such processes may improve brain function[10,13–28]. Because the brain responds to sensory inputs, particular sensory stimuli could modulate brain rhythms in a predictable and possibly therapeutic manner. Using sensory stimulation to meaningfully alter brain activity in humans, however, requires greater understanding of the spatiotemporal extent of sensory responses, mechanisms of action, and effects on pathophysiological activity throughout the brain.

Audio-visual or multi-sensory stimuli that rhythmically flicker on and off induce a steady-state evoked potential (steady-state EP), a periodic neurophysiological oscillation with similar on-off kinetics as the stimulus[29]. However, the extent and magnitude of the steady-state EP across primary sensory versus canonically non-sensory circuits is unclear. The steady-state EP has been implicated in processing of auditory stimuli such as speech[4], and its abnormality is associated with neuropsychiatric conditions such as schizophrenia[30,31] and autism[32]. This neurophysiological response has been extensively studied in humans, albeit mostly with limited spatial or temporal resolution. Most prior studies of steady-state EP used scalp electroencephalography[33–43] (EEG), where the recorded signal is non-focal and results from the complex summation of voltage changes across multiple locations, largely from superficial primary sensory circuits. In the case of auditory stimulation, the EEG-recorded steady-state EP could result from the staggered summation of signals spanning from brainstem to auditory cortex[39]. Other studies have used magnetoencephalography (MEG)[44–48], or functional magnetic resonance imaging (fMRI)[34,37,49,50] and positron emission tomography (PET)[33,38,51], which offer information about deep structures but carry limitations of spatial and/or temporal resolution. Some studies have applied the higher spatiotemporal resolution of intracranial EEG (iEEG) to more directly sample sensory circuits[52,53], but only a few have examined the effects of sensory flicker in cognitive regions implicated in disease[54–58]. The latter studies were ultimately limited by sample size, scope and depth of electrode coverage, stimulation parameters, and/or referencing methods. Thus, critical gaps in knowledge about focality and impact of sensory flicker on human brain activity have limited its application as a neurostimulation method.

The mechanisms underlying the steady-state EP remain unclear. One hypothesis posits that this response emerges from the linear superposition of individual sensory single pulse evoked potentials (single pulse EPs)[59]. Others propose that the steady-state EP results from intrinsic oscillatory circuit mechanisms, such as entrainment of active endogenous oscillations, or the resonance or preference of such circuits to oscillate at given frequencies of stimulation. While entrainment suggests that flicker could modulate functions associated with particular brain oscillations[60], the resonance hypothesis implies that flicker could preferentially target specific circuits by exploiting preferred resonant frequencies. Prior human studies of steady-state EP have offered conflicting perspectives[56,61–63] which have been hampered by sampling and methodological limitations.

Contradictory effects of sensory stimulation on pathophysiological neural activity were previously observed. Of particular interest are interictal epileptiform discharges (IEDs), which induce cognitive impairment[64,65] and versions of which are prevalent in epilepsy, AD[66,67], autism, attention deficit hyperactivity disorder (ADHD), and multiple sclerosis[68]. Relatedly, restoring gamma oscillations and the associated function of interneurons in a mouse model of AD pathology improved seizure activity and memory[13,14]. Moreover, exposing AD-model mice to gamma frequency (40 Hz) sensory flicker modulated neural firing in the medial temporal lobe (MTL) and prefrontal cortex (PFC), decreased accumulation of pathogenic peptide amyloid beta, and improved spatial memory deficits[69]. A few recent human studies indicated that flickering sensory stimuli may reduce IEDs in certain epilepsy patients[70–72], including those with focal temporal lobe epilepsy. By contrast, visual stimulation at lower frequencies around 15–25 Hz[73] can synchronize epileptogenic networks and trigger generalized seizures in genetically susceptible individuals such as 1 in 10,000 individuals in the general population and 2–14% in patients with known generalized epilepsy[73]. Overall, no study has broadly characterized the effects of varying flicker modality and frequency upon IEDs across the human brain.

To determine how multisensory flicker impacts normal and abnormal brain activity across widespread circuits, we examined the neurophysiological effects of visual and auditory flicker of multiple frequencies in 19 awake neurosurgery patients using invasive stereoelectroencephalography (SEEG), a gold standard method for recording neural activity with high spatiotemporal resolution. We recorded local field potentials (LFPs) across broad cortical regions in every patient, and additionally sampled single neuron activity in the MTL and medial PFC of a subset of patients. We report that flicker affects neural activity across widespread brain structures, including those central to multiple cognitive functions, such as the MTL and PFC. Mechanistically, our findings are consistent with a model in which flicker-induced steady-state EP emerges from resonance of circuits rather than linear superposition of single pulse EPs. Finally, we found that while flicker modestly decreases the overall rate of IEDs in focal epilepsy patients, such effects were particularly robust in locations in which LFP was more responsive to flicker, including visual cortex and MTL. Furthermore, flicker effects on IED rates depend on flicker modality, IED location, and individual subjects' seizure onset zone, implying that flicker might be tailored to distinct needs. Our results show that multisensory flicker modulates widespread brain networks and decreases pathological epileptiform neural activity in humans.

## Results

### Audio-visual flicker modulates the human MTL and PFC

We investigated the effects of flicker on human brain activity by evaluating treatment-resistant epilepsy patients undergoing pre-surgical intracranial seizure monitoring. In a first experiment, we exposed 13 participants to 10 s trials of 5.5 Hz, 40 Hz, 80 Hz, and random non-periodic visual (V), audio-visual (AV) and auditory (A) flicker, as well as baseline without stimulation (total of 13 conditions). During stimulation and baseline periods we recorded LFP from clinically targeted regions (2067 contact locations; Fig. 1A, B, and S1B). Frequencies of 5.5 Hz, 40 Hz, and 80 Hz were selected to mimic endogenous theta, slow gamma, and fast gamma brain rhythms[74], respectively, which may be relevant to cognitive processing and memory consolidation. In particular a previous study[16] suggested that modulating brain oscillations at 5.5 Hz or theta-like frequency may improve memory consolidation, while fast gamma is also thought to be involved in memory[75–77]. We compared the effects of each stimulation condition to baseline (without stimulation). We also examined neural responses to both periodic and non-periodic (random) stimuli to investigate neural responses to these different types of stimuli. As a positive control, we first confirmed that flicker modulates canonical visual processing cortices (including the pericalcarine, cuneus, lingual and lateral occipital cortices) and auditory processing cortices (including the transverse temporal gyrus and the superior temporal gyrus). We defined modulation as a statistically significant fold-change in power at the frequency of stimulation relative to baseline. We observed that 40Hz-V flicker modulated 58.1% of the contacts in visual areas, with median 5.1 fold-change in power at the frequency of stimulation relative to baseline (25th and 75th percentiles 1.5 and 19.2), compared to 15.0% of contacts in auditory areas with a median 0.7 fold-change in power (25th and 75th percentiles 0.4 and 3.8) (Fig. 1C, D). Of note, some of the temporal modulation observed with 40Hz-V stimulation may be due to early visual processing from the third visual pathway in the superior temporal sulcus[78]. In contrast, 40Hz-A stimulation modulated 18.3%

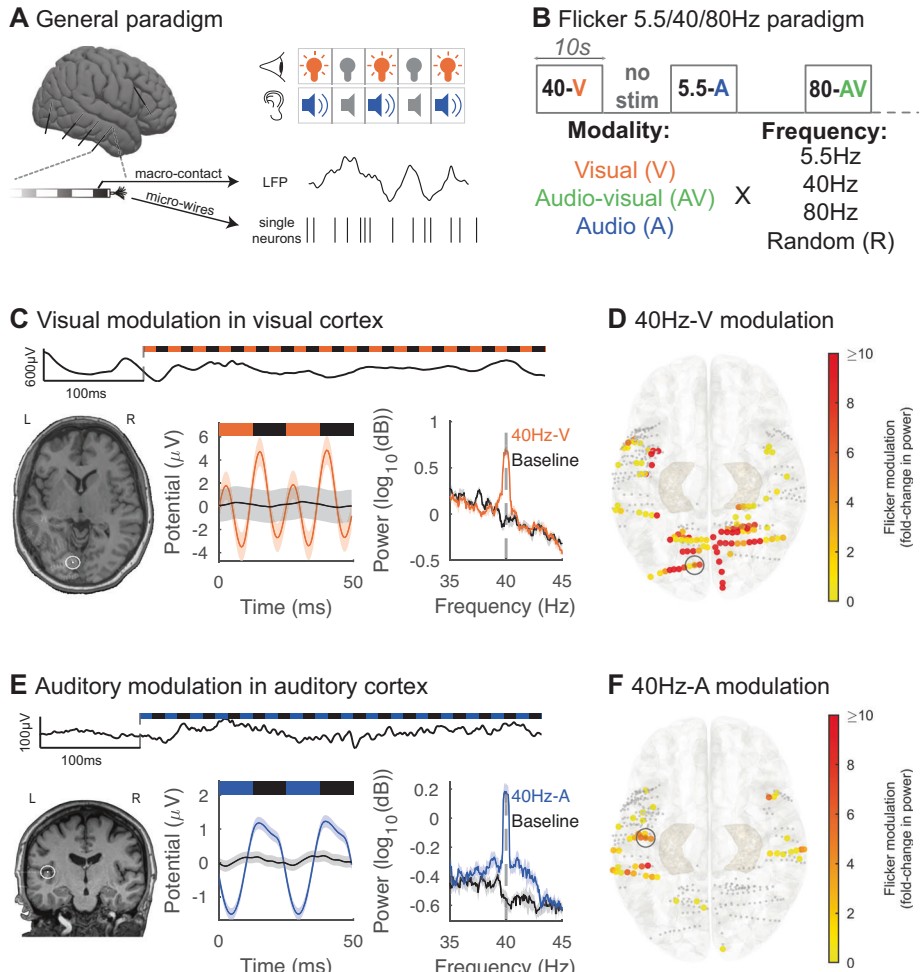

**Fig. 1 | Auditory and visual flicker induce a steady-state evoked potential in human sensory regions. A** Intracranial local field potential (LFP) and single neuron activity were recorded while we exposed subjects to visual and auditory stimulation pulses. **B** In this first paradigm, we exposed subjects to 10 s trials of visual (V, orange), audio-visual (AV, green) or auditory (A, blue) flicker at 5.5 Hz, 40 Hz, 80 Hz and random non-periodic stimuli, as well as no stimulation or baseline (total of 13 conditions). **C** Example of 40 Hz-V steady-state evoked potential (EP) in early visual area lingual gyrus in one subject. Top: raw LFP trace at the beginning of a 40 Hz-V trial. Bottom left: post-operative computed tomography (CT) scan overlaid on pre-operative magnetic resonance imaging (MRI), with contact from which results are shown highlighted with white circle. Bottom middle: LFP response to 40 Hz-V flicker (orange) and baseline (black), averaged over 2 cycles of the stimulus. Bottom right: average power spectral density (PSD) plot of 40 Hz-V flicker versus baseline. For these last two plots, lines and shaded areas indicate mean +/− standard error of the mean (SEM). **D** Response to 40Hz-V stimulation across contacts (dots) located in early visual and auditory areas, on the Montreal Neurological Institute (MNI) normalized brain (top view), with color representing fold-change in power at the frequency of stimulation, capped at 10-fold increase to best visualize this range (n = 337 channels, 12 sessions, 12 subjects). Smaller gray dots indicate channels with no significant response. 34 contacts had modulation greater than 10-fold. The contact from which results are represented in (**C**) is highlighted with a gray circle. **E** Same as in (**C**) but illustrating 40 Hz-A steady-state EP in Heschl's gyrus or transverse temporal gyrus (primary auditory area) from one subject during 40 Hz auditory stimulation (blue) or baseline (black). **F** Same as (**D**) but showing response to 40 Hz-A stimulation (n = 337 channels, 12 sessions, 12 subjects). One contact had modulation greater than the 10-fold threshold. The contact from which results are represented in (**E**) is highlighted with a gray circle. Source data are provided as a Source Data file.

of contacts in auditory regions with median 1.6-fold increase in power (25th and 75th percentiles 0.6 and 3.9), compared to 3.2% of contacts in visual areas with median 0.4 fold-change (25th and 75th percentiles 0.3 and 0.5) (Fig. 1E, F). 5.5 Hz and 80 Hz stimulation conditions led to similar modulation of respective sensory regions (Fig. S2). We note that in some instances, sensory flicker induced a double-peak per stimulation cycle (likely representing neural responses to both the onset and offset of stimulus pulses) or other more complex responses (Fig. 1C). In some instances, sensory modulation across patients yielded asymmetrically stronger responses from one hemisphere over the other (Fig. 1D, F). Since few patients were sampled equally from both hemispheres, asymmetric responses may have resulted from heterogenous brain sampling. Overall, these results confirmed that flicker stimulation effectively engages canonical sensory regions.

We then investigated whether sensory flicker modulates activity in higher cognitive regions, specifically the MTL and PFC, which are frequently sampled in subjects with focal-onset epilepsy. We found that flicker consistently alters LFP at locations within these structures, with increased spectral power at the frequency of stimulation compared to baseline (Fig. 2A, B). Across 326 MTL and 467 PFC contacts from 13 subjects, we found 40Hz-AV flicker significantly modulated 13.8% and 8.1% of contacts, respectively, with a median 1.1-fold increase (25th and 75th percentiles 0.5 and 1.7) and 0.4-fold increase (25th and 75th percentiles 0.3 and 0.7) in power relative to baseline, respectively for the MTL and PFC (Fig. 2C, D). These results are unlikely to be explained by volume conduction of electrical potentials from nearby sensory-processing areas or by artifacts from our stimulation device, as we controlled for both by using off-line Laplacian re-referencing (see Methods) and by showing absence or reduction of a response

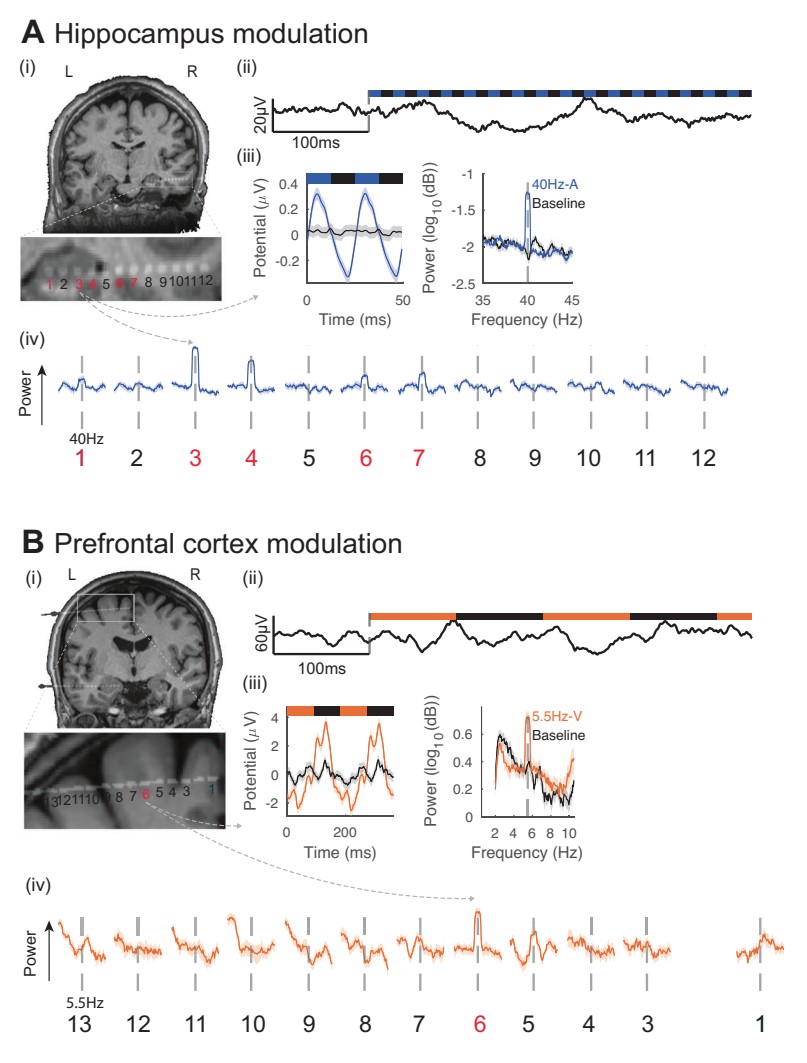

**A** Hippocampus modulation

**B** Prefrontal cortex modulation

**C** 40Hz-AV MTL and PFC modulation

**D** 40Hz-AV MTL and PFC modulation

under a relative occluded stimulation (Fig. S1A). Of note, some of the contacts located in temporal lobe white matter (not MTL) lateral to the hippocampus exhibited responses that may be due to sampling of optic radiations[79], which are involved in early visual processing. Significant modulation was often highly localized to one or a few contacts on a recording probe (Fig. 2A, B). As LFPs are considered to represent synchronized currents from organized dendritic inputs[80], these results show that the MTL and PFC are modulated by sensory flicker despite being canonically thought to be beyond primary sensory processing. We also found modulation in response to 5.5 Hz and 80 Hz audio-visual stimulation in the MTL and PFC (Fig. S3A). Again, asymmetric

responses likely reflect heterogeneous electrode placement across subjects (Fig. 2C). To contrast the responses to periodic versus random flicker stimulation, we compared the specificity of modulation at the frequency of stimulation of 40 Hz to random conditions. Both periodic and random flicker have frequency components (the frequencies at which the stimuli turn on and off), the difference being that periodic flicker has a very narrow frequency band while random flicker has a wide frequency band. Thus, we would expect that both conditions would increase power in the frequency of the flickering stimulus: a narrow band for periodic flicker and a wide band for random flicker. Indeed, random, non-periodic stimulation induced increases in LFP

**Fig. 2 | Audio-visual flicker induces a steady-state evoked potential in the human medial temporal lobe and prefrontal cortex. A** Example of 40 Hz-auditory (A) steady-state evoked potential (EP) in the hippocampus (HPC). (i) location of a depth electrode with contacts numbered from deep to superficial (zoomed-in below), in one subject, overlaid on pre-operative MRI; contacts 1–5 are in or near the hippocampus. (ii) Example local field potential (LFP) trace for the beginning of a 40Hz-A trial. (iii) For the same contact, averaged evoked potential over 2 cycles of the stimulus (left), averaged power spectral density (PSD) during 40Hz-A flicker (blue) and baseline (black, right). (iv) Averaged PSD for each contact from the depth electrode during 40Hz-A flicker, zoomed-in to show frequency of stimulation +/− 5 Hz (solid lines and shaded areas: mean +/− SEM), showing evoked responses in red contacts 3, 4, in the hippocampus, and weaker response in contacts 1, 6 and 7. **B** Same as (**A**), for a depth electrode in the superior prefrontal cortex (PFC) in a different subject, during 5.5 Hz-visual (V) flicker (orange). **C** Electrode contacts

(dots) in the medial temporal lobe (MTL) and PFC that were modulated by 40 Hz-audiovisual (AV) flicker, on a MNI normalized brain (top view), with color representing fold-change in power at the frequency of stimulation, capped at 2-fold increase to best visualize this range ($n$ = 793 contacts, 13 sessions, 13 subjects). Gray dots: contacts with no significant response. **D** Middle: fold-change in power (capped at 2-fold change) at 40 Hz during 40 Hz-AV flicker relative to baseline, for contacts with significant modulation; percent of electrodes showing significant steady-state EP above ($n$ = 326 and 467 contacts, 13 sessions and subjects). Black open circles: medians, vertical lines: whisker plots, filled dots: each contact. 9 contacts had modulation higher than capped 2-fold change in the MTL, and 4 contacts in the PFC. Left and right: example power spectral densities during 40 Hz-AV flicker (green) or baseline (black) in the MTL and PFC, respectively. Lines and shaded areas: mean +/− SEM. Examples highlighted with red circles in the violin plots. Source data are provided as a Source Data file.

power in broad frequency ranges when a strong sensory response was present, while periodic stimulation induced a narrow band increase in power at the frequency of stimulation (Fig. S4). We also evaluated whether spiking activities of single neurons in higher cognitive regions, specifically the hippocampus and cingulate gyrus, were modulated by flicker. We found evidence for modulation of some single and multi-units in both regions (Fig. S5), although the small number of recorded units (21 with high enough spiking rate for analysis) did not support any strong conclusions.

## Audio-visual flicker induces steady-state EPs across widespread functional networks

Next, we studied the extent of flicker modulation across the brain, and how regions respond to specific modalities and frequencies of stimulation, which may inform which regions can be targeted with this approach. Overall, audio-visual flicker produced the broadest responses, i.e., more contacts with significant steady-state evoked potential, across all frequencies tested, followed by visual flicker and auditory flicker (chi-square statistic of the difference between proportions of modulated contacts 46.9, $p$ = 6.6 × 10$^{-11}$, df = 2, 3 proportions, 2067 channels included; Fig. 3A, top). With respect to frequencies of stimulation tested, more contacts exhibited a steady-state EP in response to 40 Hz, than to 5.5 Hz or 80 Hz stimulation (chi-square statistic of the difference between proportions of modulated contacts 47.8, $p$ = 4.1 × 10$^{-11}$, df = 2, 3 proportions, 2067 channels included; Fig. 3A, center). For locations responding to both visual and auditory flicker, the majority (64.8%) showed preferential responses to the same stimulation frequency (Fig. 3A, bottom), suggesting that brain regions are sensitive to given frequencies of stimulation, irrespective of the modality. We determined the relative strength (fold-change in power) of steady-state EP in response to flicker by spatial distribution, modality, and frequency (Fig. 3B). As expected, we observed a strong response in the occipital region for conditions involving the visual modality, as well as moderate responses of the parietal, temporal and prefrontal regions; the auditory-only condition mainly affected temporal and prefrontal regions; the 40 Hz condition appeared to broadly impact most regions, particularly when using combined visual and auditory modalities. We also determined the strength of 40 Hz-AV flicker-induced steady-state EP by functional networks (Fig. 3C) that were previously defined by fMRI resting state functional connectivity across 1,000 healthy subjects[81]. More than half (56.3%) of the contact locations in or near the visual network showed significant modulation to flicker with some locations having a more than 10-fold increase in power. More notably, flicker affected subsets of contact locations throughout networks not thought to be primarily involved in sensory processing, with 19.9%, 14.8%, 15.6%, 12.5% and 7.9% of the contacts showing significant steady-state EP in the ventral-attention, dorsal-attention, default, limbic, and frontoparietal networks, respectively. 5.5 Hz and 80 Hz audiovisual stimulation also led to modulation in these same networks (Fig. S3B).

## The steady-state EP does not result from linear superposition of single pulse EPs

Next, we tested potential mechanisms by which flicker induces a steady-state EP. Depending on the mechanisms involved, sensory flicker could modulate brain functions associated with endogenous oscillations, or flicker frequency could be used to optimize targeting of modulation to regions of interest. One common hypothesis is that the steady-state EP results from the linear superposition of single pulse EPs[59,82,83] (Fig. 4A). This contrasts with other proposed mechanisms, in which the steady-state EP results from intrinsic oscillatory properties of circuits that exhibit greater responses to specific stimulation frequencies. We first tested the linear superposition mechanism. Specifically, this mechanism predicts that (1) a region showing a single pulse EP should also show a steady-state EP, (2) the amplitude of the steady-state EP should be proportional to that of the single pulse EP such that a region showing a strong sensory response to single pulses should show a correspondingly strong response to flicker, (3) the steady-state EP amplitude should be approximated by simulating the linear superposition of single pulse EPs, (4) the amplitude of the response to flicker should decrease as the stimulation frequency increases, due to the low-pass filter properties of neurons and circuits[84], and (5) there should be no interaction between steady-state EP and endogenous oscillations, but rather a simple superposition or co-existence of the two.

To test the superposition hypothesis, we ran an additional experiment in a subset of 6 subjects (total of 1025 recording contact locations), where we exposed them to single pulses of visual, audio-visual, and auditory modalities (Figs. 4B, S1B and S6A). We then quantified, for each contact, whether there was a single pulse EP and the magnitude of that response (see Methods). As expected, we found stronger responses in the occipital region with visual modality, and in the temporal region with auditory modality (Fig. S6B). Surprisingly, we found widespread single-pulse EPs beyond sensory regions. Moreover, we found that among locations showing any sensory response, 45.7–54.9% responded to single pulses only (45.7%, 46.1% and 54.9% for visual, audio-visual and auditory modalities respectively), while 17.4–25.8% showed response to flicker only (21.1%, 17.4% and 25.8%), and 19.3–36.5% responded to both single pulse and flicker (33.2%, 36.5% and 19.3%) (Figs. 4C and S6C). These results indicate a discrepancy between single pulse versus flicker responses for most regions sampled, which is inconsistent with the superposition hypothesis and may imply involvement of additional mechanisms, such as sensory adaptation[85] or other modulatory dynamics specific to circuits involved in the processing of the stimulus.

We then considered if the superposition mechanism specifically explained responses of the subset of locations showing both a significant steady-state EP and single-pulse EP, rather than all recorded responses. If true, we would expect proportional amplitudes of the steady-state and single-pulse EPs in this subset of locations. Instead, we observed that the normalized log-scaled amplitudes of the flicker

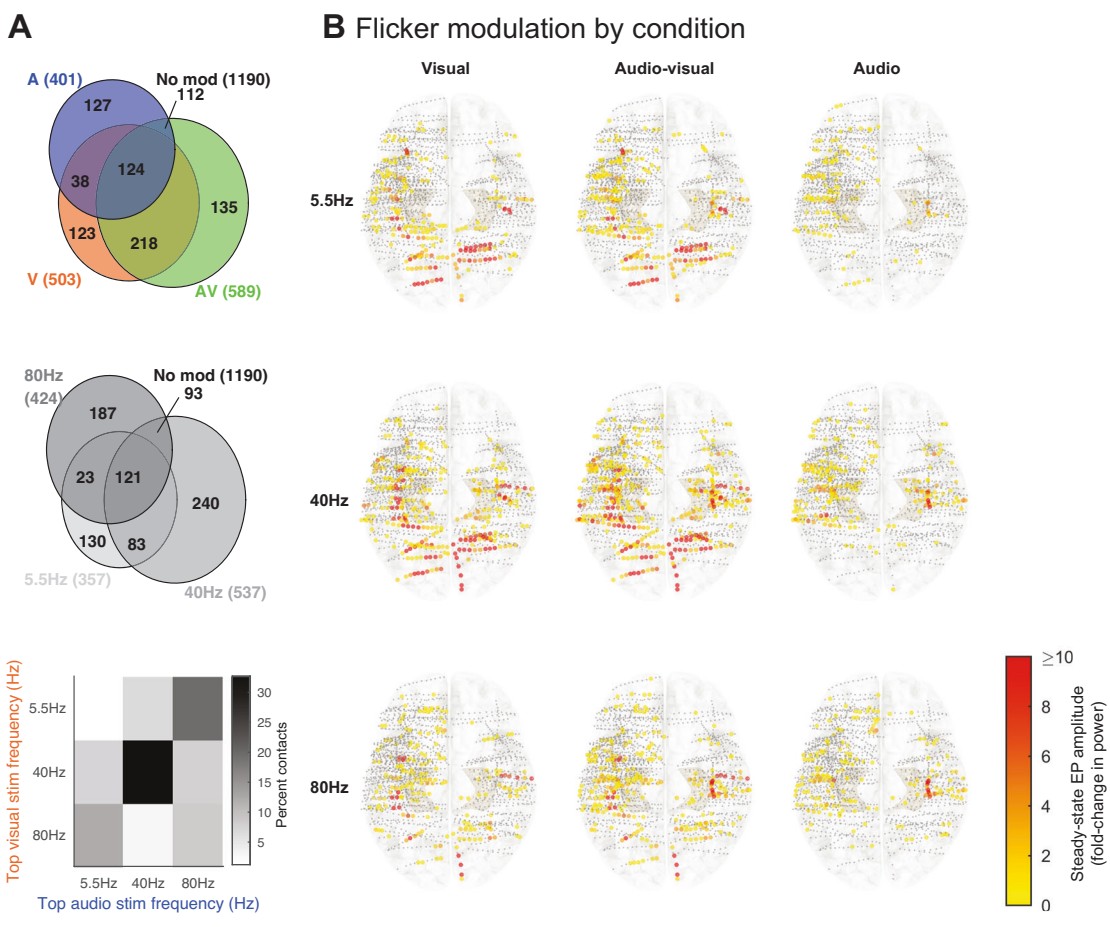

**A**

**B** Flicker modulation by condition

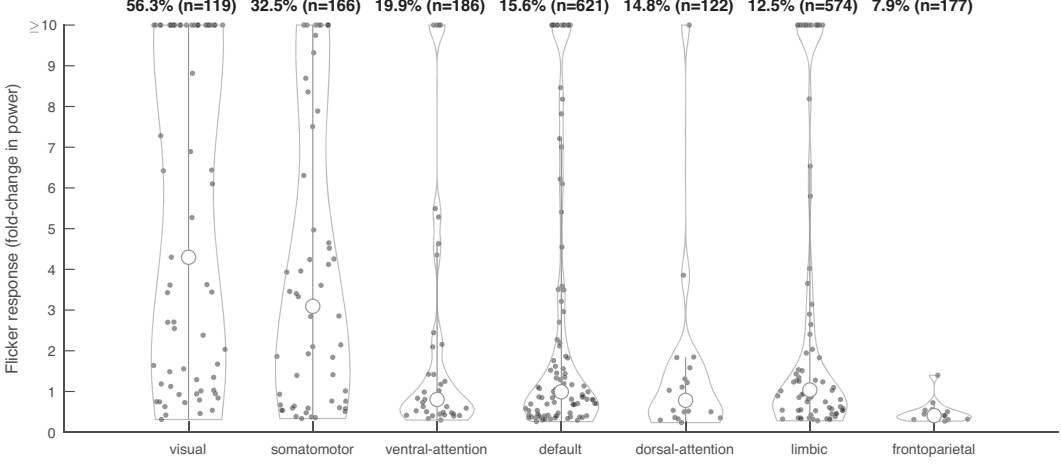

**C** 40Hz-AV modulation by functional network

response and single-pulse EP were significantly different (paired-sample, two-sided *t*-test, *p* = 0, confidence interval (ci) = [0.6270 0.7830], t-statistic (tstat) = 17.7564, degrees of freedom (df) = 530, standard deviation (sd) = 0.9150), and that the significance of these responses also were different (paired-sample, two-sided *t*-test, *p* = 6.3022 × 10$^{-8}$, ci = [−0.0050 −0.0024], tstat = −5.4882, df = 530, sd = 0.0155; Fig. 4D). Most recorded locations showed a trend for a stronger single-pulse EP compared to steady-state EP, in contrast to the prediction of the superposition hypothesis.

One possible explanation for stronger single-pulse versus steady-state EP is destructive interference, in which a peak of one pulse EP coincides with a trough of the previous pulse EP, resulting in an overall attenuated response in the case of steady-state EP. Likewise, constructive interference of single pulse EPs might explain higher amplitude modulation at given frequencies of stimulation, such as suggested by scalp EEG studies of the 40 Hz-A response[39]. We tested, for each recording site showing significant 40 Hz steady-state EP, whether we could artificially generate the expected flicker response

**Fig. 3 | Flicker steady-state evoked potential across the brain. A** Top: Venn diagram showing proportion of contacts (*n* = 2067 contacts, 13 sessions, 13 subjects) with significant steady-state evoked potential (EP) to visual (V, orange), audio-visual (AV, green) and auditory (A, blue) flicker; absolute number of contacts are also shown. Center: Venn diagram showing significant responses to different flicker frequencies (5.5 Hz- light gray, 40 Hz- darker gray, 80 Hz- dark gray). Bottom: top stimulation frequency for each modality for contacts that responded to both visual and auditory flicker. Most contacts showed a preference for the same stimulation frequency when stimulated with either modality. **B** Approximate location of each contact (dots) and associated amplitude of steady-state EP, plotted on Montreal Neurological Institute (MNI) normalized brain (top view), for each of 9 conditions: 5.5 Hz, 40 Hz, 80 Hz stimulation frequencies at visual (V), audio-visual (AV) and

auditory (A) modalities. Color of larger dots indicates power fold-change in channels with significant steady-state EP, from yellow to red (0–10-fold or more increase in power). Smaller gray dots indicate no significant response. **C** Distribution of 40Hz-AV flicker steady-state EP across all contacts showing significant modulation from all subjects, categorized by functional networks (as previously defined by resting state functional connectivity characterized across 1000 healthy subjects[81]). Percent of contacts in that network with significant responses, with absolute number of contacts localized to those networks in parentheses (*n* = 1965 contacts, 13 sessions, 13 subjects). Open circles represent medians of the distributions, vertical lines indicate whisker plots, filled dots indicate each significant contact. Source data are provided as a Source Data file.

based on the single pulse EP shape and amplitude for each of those contacts, as predicted by the superposition mechanism (see Methods). Such a simulation should account for destructive or constructive interference and thus explain a low flicker amplitude response compared to single-pulse EP. We found that overall, out of recording sites showing significant 40 Hz modulation, only a minority showed significant modulation in the simulated data (24.5%, 25.1% and 18.4%, respectively, in the visual, audio-visual and auditory conditions), and at a lower amplitude (0.2 versus 1.1, 0.2 versus 1.2, 0.3 versus 0.6 median fold-changes, respectively for visual, audio-visual and auditory conditions). The distribution of amplitudes of steady-state EP was significantly higher in the real data compared to the simulated data (Fig. 4E, left; paired-sample, two-sided t-test of the real vs simulated data non-capped distributions for the visual condition $p = 0.0093$, ci = [4.9421 34.7771], tstat = 2.6246, df = 207, sd = 109.1273, for the audio-visual condition $p = 0.0081$, ci = [4.9576 32.8400], tstat = 2.6703, df = 242, sd = 110.3258, for the auditory condition $p = 2.5473 \times 10^{-6}$, ci = [0.7363 1.7096], tstat = 4.9847, df = 102, sd = 2.4900), and those values were significantly different from each other overall (Fig. 4E, right; paired, two-sided *t*-test, $p = 1.6655 \times 10^{-4}$, ci = [7.6969 24.2496], tstat = 3.7910, df = 553, sd = 99.1734). This result provides additional evidence against the linear superposition hypothesis and suggests that destructive interference does not explain the overall lower amplitude of the flicker response compared to single-pulse EP. Overall, our data does not fit the expected outcomes from the linear superstition hypothesis, making this an unlikely underlying mechanism for flicker modulation.

## Resonance of long-range circuits to specific flicker frequencies

Since the linear superposition hypothesis did not explain the observed steady-state EP, we hypothesized that the steady-state EP might arise from intrinsic oscillatory properties of circuits involved in processing of the stimulus. Such a mechanism predicts that steady-state EPs may emerge in the absence of single pulse EPs, and that the flicker response could show resonance[28,84] of stimulated circuits or entrainment[4,18,61,86–88] of active endogenous oscillations due to intrinsic oscillatory circuit properties. We defined resonance as the preferential response of the stimulated circuit (from early sensory input structures to recorded region) to specific frequency bands of stimulation, which is presumed to result from underlying neuronal features and synaptic connectivity[84]. We predicted that resonance would manifest as (1) a stronger steady-state EP at a subset of flicker frequencies tested and (2) a stronger phase-locking value (PLV) between stimulus and LFP at similar frequencies. While the term entrainment has been used in a variety of ways, we defined entrainment of endogenous oscillations narrowly[4] here as the ability of a repetitive stimulus to modulate an active endogenous oscillation detected at a recording site. We expected that entrainment would manifest as (1) a stronger change in power and (2) a stronger phase-locking with stimulation at a frequency close to that of a detected active endogenous oscillation.

To test these predictions, we exposed a separate subset of 8 subjects (total of 1339 contact locations across 11 sessions) to visual or

auditory flicker at 26 different frequencies spanning the 5.5–80 Hz range in about 3 Hz intervals (Figs. 5A and S1B), as well as random flicker and baseline (no stimulation), and estimated the amplitude of the steady-state EP resulting from each stimulation frequency. First, we observed that most recorded locations with a steady-state EP exhibit a preferential response to specific frequencies of stimulation (Fig. 5B, C), in line with intrinsic circuit oscillatory mechanisms. We found that most contact locations across sessions were modulated, with 74.8% and 84.3% of all recorded contacts showing significant fold-change in power or PLV to the stimulus, respectively, for at least one stimulation frequency (Fig. 5C). Different locations exhibited strong responses to varying stimulation frequencies over the entire tested range of 5.5–80 Hz. Moreover, 14.7% and 21.4% of contacts showed significant fold-change in power or PLV (median of 1.8 and 0.284, range of 0.02–545.3 and 0.04–0.97, 25th percentile of 0.9 and 0.192, 75th percentile of 5 and 0.468, respectively) to more than 6 of the stimulation frequencies tested. Of these only 2.9% and 3.7% showed a preference or strongest response to the lowest frequency of stimulation and otherwise most showed a preference for a variety of frequencies. Other contacts showed a preference for fewer (1–6) stimulation frequencies (60.1% and 62.9% of contacts showing significant fold-change or PLV, respectively), albeit their modulation values were lower (median of 0.6 and 0.134, range of 0.2–9.3 and 0.045–0.344, 25th percentile of 0.4 and 0.115, 75th percentile of 0.7 and 0.156, respectively).

We also explored whether specific recorded regions might show concordant preferred responses to specific stimulation frequencies, among channels that showed significant fold-change at the frequency of stimulation, for more than six of the stimulation frequencies tested (Fig. S7). We did not observe a clear clustering of stimulation frequency preference by recorded brain region, regardless of the modality of stimulation (visual or auditory) used. Overall, most contact locations showed preferential response to stimulation at a given frequency, supporting an intrinsic circuit oscillatory mechanism. Furthermore, we predicted that under this mechanism, flicker and single pulse responses do not necessarily match, both in terms of presence and amplitudes. We already showed that a subset of contact locations (17.4–25.8%) had significant flicker responses without exhibiting single-pulse EP, indicating that single-pulse EP may not be necessary for steady-state EP (Fig. 4C). We also found that the amplitudes of steady-state EP and single pulse EP are often different, with a tendency for stronger single-pulse EP (Fig. 4D).

We further examined whether flicker-induced oscillatory activity persisted beyond the offset of stimulation, which would be another indication towards an intrinsic oscillatory mechanism of sensory flicker modulation[62,89,90]. Using a method developed by Lerousseau et al.[56], we detected the onset and offset of oscillatory response to sensory flicker in responding contacts (Fig. S8A). In strongly modulated contacts, we found examples of both persistent oscillatory activity by more than one cycle beyond where we would expect it to terminate based on the delay in sensory response (Fig. S8B, left), as well as absence of persistent oscillatory activity (Fig. S8B, right).

**A** Hypothesized mechanism

**B** Single-pulse paradigm

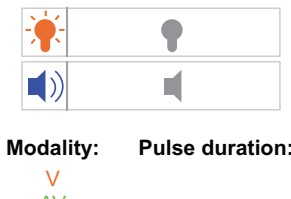

Modality:   Pulse duration:

V
AV    X    12.5ms
A
Occluded AV

**C** Single pulse vs flicker response

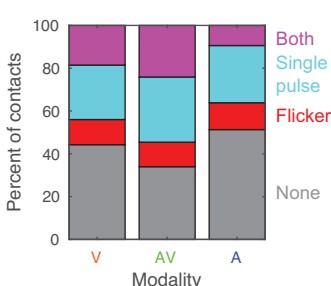

**D** Amplitudes of single pulse vs flicker responses

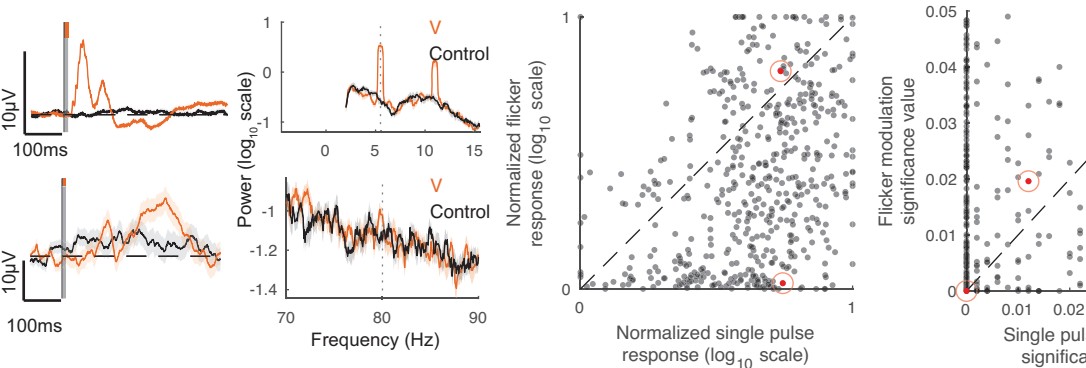

**E** Simulated vs measured flicker response

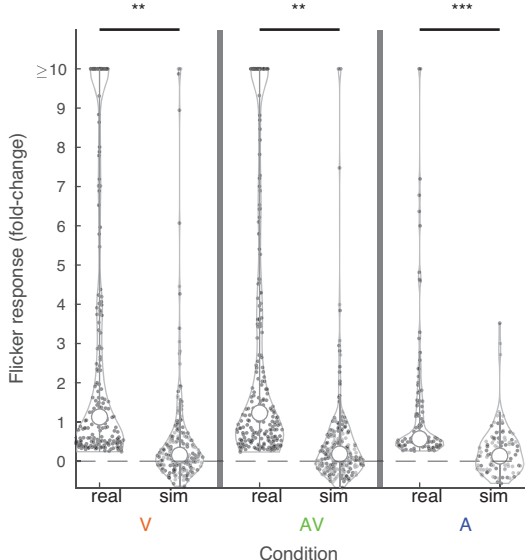

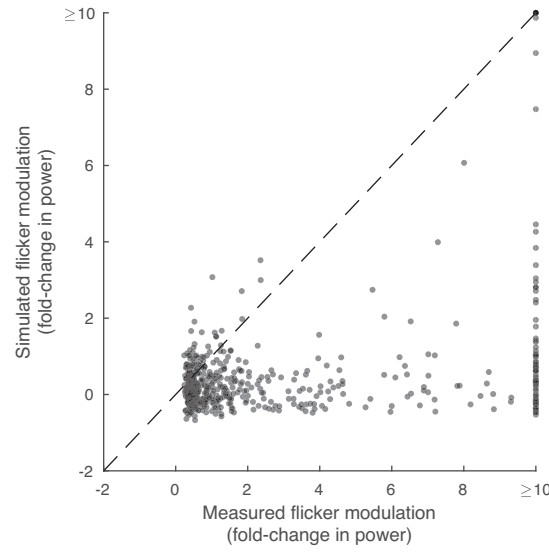

Overall, evidence of persistent oscillatory activity further suggests intrinsic oscillatory mechanisms underly the flicker response.

Next, we probed whether the observed preference for given frequencies of stimulation was related to active endogenous oscillations detected at recording sites, which would suggest entrainment of such oscillations by flicker. If flicker responses result from entrainment of active endogenous oscillations, then the flicker steady-state EP should be strongest at frequencies that are closest to those oscillations, a phenomenon illustrated by one dimension of the so-called Arnold tongue[4]. To test this prediction, we compared the response to flicker stimulation at frequencies spanning the 5.5Hz-80Hz stimulation range to baseline active endogenous frequencies (see Methods). We found that across recorded contact locations, many showed strong endogenous oscillations in the alpha (~10 Hz) and beta (~20 Hz) ranges, with some also exhibiting higher frequency oscillations (Fig. 5D). We hypothesized that stimulation frequencies eliciting maximal fold-change at the stimulation frequency, or top stimulation frequencies (examples illustrated with dashed colored line in Fig. 5B), may be similar to strong active endogenous oscillations. We focused on contacts with significant fold-change in power for more than 6 out of the

**Fig. 4 | Flicker modulation does not result from linear superposition of single pulse evoked potentials. A** Schematic of the hypothesis that the steady state EP results from linear superposition of single pulse EPs: 40 Hz steady state EP (black) is hypothesized to result from the response to a single visual pulse (orange) that is repeated every 25 ms (transparent gray) and linearly summed. **B** 6 subjects were exposed to single 12.5 ms pulses in the visual (orange), audio-visual (green), and auditory (blue) modalities or relative occluded flicker (with sleep mask and ear-plugs) as control. **C** Percent of contacts showing response to flicker-only (red), single pulse-only (cyan), both flicker and single pulses (purple), and no response (gray), with the visual, audiovisual, and auditory modality, respectively. **D** Left: Example contacts that responded strongly to both (top) single visual pulses (left) and visual flicker (right) or more strongly to single pulses (bottom) during stimulation relative to control (black). Single-pulse EP (control is relative occluded audio-visual) and power spectral density (PSD) plots (control is baseline) shown for each. Lines and shaded areas: mean +/− SEM. Middle: Steady-state EP versus single-pulse

EP amplitude, normalized by subject and stimulation modality; each dot represents one contact's responses for a given modality; contacts with both significant steady-state EP and single-pulse EP were included ($n = 319$ contacts, 6 subjects). Red dots indicate examples on left. Right: Significance values of flicker versus single pulse response. **E** Left: Steady-state EP fold-change in power at 40 Hz in the visual, audio-visual, and auditory modalities, for real and simulated data across contacts (two-sided $t$-test; visual condition $p = 0.0093$, audio-visual condition $p = 0.0081$, auditory condition $p = 2.5473 \times 10^{-6}$; $p$-values are uncorrected for multiple comparisons and are lower than Bonferroni correction for 3 comparisons; $n = 554$ contacts, 6 subjects.) Only contacts showing significant flicker modulation in the real data were included. **$p < 0.01$, ***$p < 0.001$, open circles: medians, vertical lines: whisker plot, dots: each contact. Right: for those same contacts, amplitude of flicker steady-state EPs calculated using real data ($x$-axis) versus using simulated data ($y$-axis). Dots: one contact and modality. Source data are provided as a Source Data file.

26 tested flicker frequencies because these locations tended to show clear preference for given frequency ranges, which may overlap with endogenous oscillations. Overall, among contact locations showing both clear baseline endogenous oscillations and significant steady-state EP for more than 6 out of the 26 tested flicker frequencies, we found only a minority (15.1%) of locations having their top stimulation frequency within 5 Hz of an endogenous frequency (Fig. 5E; paired, two-sided $t$-test of endogenous and optimal stimulation frequencies, $p = 0$, ci = [−33.6 −27.8], tstat = −21.0, df = 191, sd = 20.2). These results show that while flicker may entrain baseline endogenous oscillations in specific recorded regions, this does not explain most steady-state EP responses. Overall, our data points to an intrinsic oscillatory circuit mechanism of flicker-oscillatory response, most likely mediated by resonance of stimulated circuits spanning from early sensory input structures to recorded structures.

## Flicker decreases IED rate differentially in focal seizure networks

Having established that flicker modulates widespread brain regions in humans, including deeper cortical structures and cognitive areas, we next evaluated whether flicker modulation alters the frequency of IEDs in focal epilepsy patients. Of note, our study population did not include generalized epilepsies (since such patients do not typically undergo intracranial seizure mapping procedures) and also excluded patients preoperatively found to be susceptible to photic-induced electrographic seizure activity (see Methods). Using a clinically-validated automated IED detection algorithm[91], we identified and quantified the proportion of IEDs occurring during 10 s trials of stimulation (Fig. 6A) and time-matched no-stimulation segments, in sessions from both Flicker 5.5/40/80 Hz- and Flicker 5.5–80 Hz-range paradigms (see Methods). Because the outcome is IED count, and to account for within-and between-subject heterogeneities in baseline IED rates, we used a Poisson generalized linear mixed effects model to quantify the effects of flicker upon IED rates.

First, we observed that across conditions tested, sensory flicker reduced the whole-brain rate of IEDs in our subject group, by 3.0% compared to baseline ($p = 2.4 \times 10^{-5}$, ci = [−4.5 −1.6], df = 10854; Fig. 6B left, S9A). This indicates that sensory flicker may exert modest whole-brain anti-epileptic effects in patients with focal epilepsy. Specific sensory flicker conditions with significant anti-IED effects at the whole-brain level highlighted gamma frequencies: 40 Hz-A, 66 Hz-A, 69 Hz-V, and 75 Hz-A conditions (−22.1%, −16.8%, −21.6%, $p = 1.62 \times 10^{-2}$, $2.71 \times 10^{-2}$, $1.65 \times 10^{-2}$, ci = [−43.7 −3.8], [−34.1 −1.8], [−42.5 −3.7], df = 114, 96, 116; Fig. S9C). While we observed lower IED rates during 40 Hz-A flicker in one paradigm, this was not replicated in another paradigm in which instead 40Hz-V flicker reduced IED rate (Fig. S9B, C). These observations are unlikely to result from experimental artifact. Indeed, there is no reason to predict condition-related biases in IED detection. If sensory-evoked potentials could theoretically contribute false positive detection of IEDs, we would

instead observe the opposite—a significant IED increase caused by sensory flicker.

We likewise observed that the magnitude of sensory flicker modulation (i.e., how strongly sensory flicker modulated neural activity at the frequency of flicker) of particular locations predicted the magnitude of IED reduction at those locations. Specifically, flicker induced a greater IED reduction in those channels that were strongly modulated (high-mod, >1.5 fold-change in power) by the respective conditions at −15.6% ($p = 2.9 \times 10^{-4}$, ci = [−25.0 −4.1], df = 8840), compared to channels that were weakly (<1.5 fold-change in power) or not significantly modulated at −2.7% ($p = 2.8 \times 10^{-4}$, ci = [−6.9 −1.2], df = 139616), with a significant difference between the two groups ($p = 3.5 \times 10^{-3}$; Fig. 6B, middle).

Next, we examined the interaction of flicker modality and IED reduction with respect to anatomical location of IEDs. Including all subjects, we found that visual and audiovisual, but not auditory, flicker robustly decreased IED rate in early visual regions by an average of 35.8% and 27.7%, respectively ($p = 2.6 \times 10^{-6}$ and $2.1 \times 10^{-4}$, ci = [−54.2 −19.5] and [−45.3 −12.2], df = 4342 and 1676, respectively; Fig. 6B, right). Notably, visual and audiovisual flicker decreased IEDs by 5.7% and 18.8% on average in the MTL ($p = 1.8 \times 10^{-3}$ and $1.6 \times 10^{-8}$, ci = [−9.4 −2.1] and [−26.1 −12.0], df = 4342 and 1676), a region not canonically considered to be directly involved in primary sensory processing. Auditory flicker did not lead to significant changes in IED in any of the regions of interest, including early visual, auditory regions, MTL, or PFC. We observed increases in IED rate during audiovisual flicker in auditory regions and PFC (17.6% and 23.1%, $p = 4.5 \times 10^{-3}$ and $3.7 \times 10^{-8}$, ci = [5.8 27.9] and [15.6 29.9], df = 1676 and 1676, respectively).

We then specifically probed how changes in IED rate were related to each individual subject's general region of seizure onset zone (SOZ), defined by anatomico-clinico-physiological correlation and used to guide subsequent epilepsy surgery by the clinical team. Changes in IED rate within this region would carry inherently greater clinical relevance compared to changes in IED rate in other regions. We used SOZ to classify patients as either temporal lobe epilepsy (TLE, $n = 14$) or frontal lobe epilepsy (FLE, $n = 3$) patients, excluding 2 patients who did not fulfill these classifications. In TLE subjects, we found visual, audiovisual and auditory flicker significantly decreased mean IED rates by 6.5%, 21.5%, and 5.5% in the MTL ($p = 4.6 \times 10^{-3}$, $5.3 \times 10^{-9}$, and $1.8 \times 10^{-2}$, ci = [−11.3 −2.0], [−29.7 −13.8] and [−10.2 −0.9], df = 3476, 1316 and 2852, respectively; Fig. S9D, left). Notably this group showed significant increases in IED rate in auditory regions and PFC, outside the region of the SOZ of each subject, during audiovisual flicker (16.4% and 30.6%, $p = 1.0 \times 10^{-2}$ and $2.7 \times 10^{-8}$, ci = [4.1 27.1] and [21.1 38.9], df = 1316 and 1316, respectively). By contrast, in FLE subjects, we found auditory flicker significantly decreased IED rate while audiovisual flicker significantly increased IED rate in PFC (−12.0% and 15.1%, $p = 1.1 \times 10^{-2}$ and $2.2 \times 10^{-2}$, ci = [−22.2 −2.7] and [2.3 26.2], df = 778 and

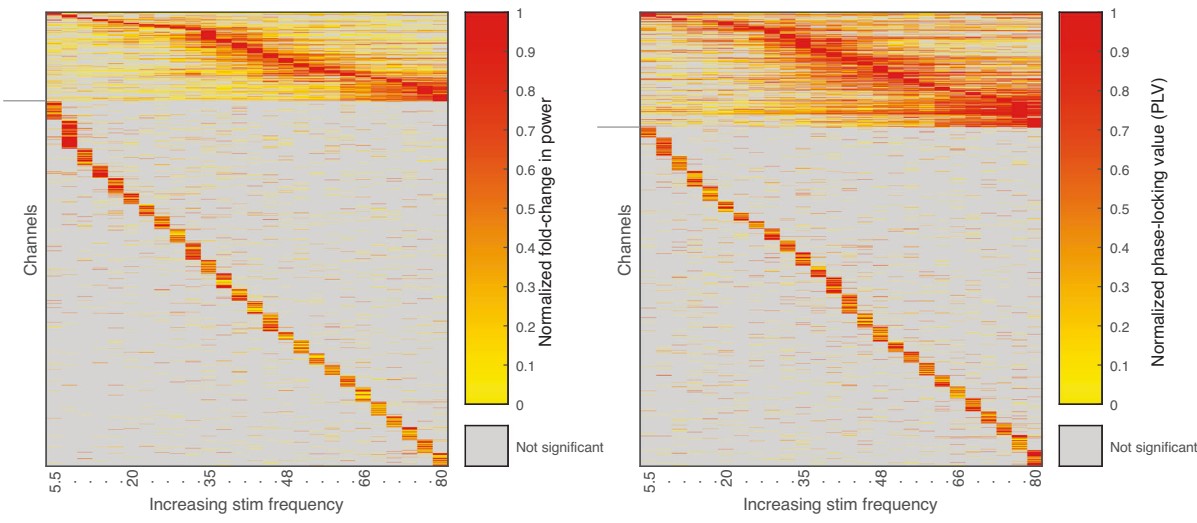

**A** Flicker 5.5-80Hz range paradigm

**B** Examples of preference for stimulation frequency

**C** Preference for stimulation frequency

**D** Endogenous oscillations

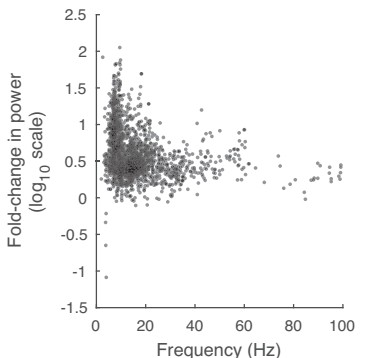

**E** Preferred stimulation frequency vs endogenous oscillations

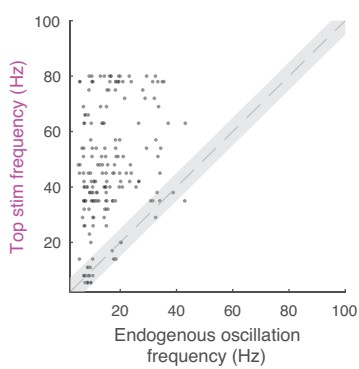

238, respectively; Fig. S9D, right), with no significant changes in MTL or other circuits assessed.

Overall, these results support the safety of sensory flicker in select focal epilepsy patients. Even brief, 10 s bouts of sensory flicker reduced IEDs across the brain, a biomarker of brain pathophysiology. Moreover, detailed examination indicated that specific effects of flicker stimulation upon IEDs varies based upon sensory modality, functional anatomic location, and location of seizure onset. In TLE patients, visual and audiovisual modality stimulation conditions clearly reduced IEDs in visual and MTL regions. In FLE patients, auditory stimulation appeared to yield a modest selective IED reduction in the PFC, while audiovisual stimulation induced an increase in IED rate in the PFC. Together, these result show that even brief exposure of such patients to sensory flicker reduces a pathophysiological biomarker – IEDs – in a

**Fig. 5 | Flicker response is dependent on intrinsic circuit properties. A** In the Flicker 5.5–80 Hz range paradigm, 8 subjects (11 sessions) were exposed to either visual (V, orange) or auditory (B, blue) modalities at 26 different frequencies spanning 5.5–80 Hz, random non-periodic flicker, and baseline (no stimulation). **B** Example contacts showing endogenous oscillations and response to stimulation frequencies. Top: power spectral density (PSD) during stimulation at each of 26 flicker frequencies, showing power values at the stimulation frequency +/− 1 Hz overlaid on the average baseline PSD (black) and aperiodic fit (1/f, gray). Lines and shading: mean +/− SEM. Bottom: fold-change in power (solid line) and phase-locking value (PLV, dotted line) for each stimulation condition. Vertical dashed colored line: stimulation frequency leading to maximal modulation, vertical dashed gray line: frequency of detected endogenous oscillation closest to top stimulation frequency, solid discs: significant fold-change, solid diamonds: significant PLV. **C** Normalized fold-change in power (left) at the frequency of stimulation and PLV (right), for each channel (rows) and frequency of stimulation (columns),

normalized across stimulation frequencies. Some channels are repeated for subjects who underwent both the visual and auditory versions of the Flicker 5.5–80 Hz range paradigm (see Table S4). Channels with significant modulation to more than 6 stimulation frequencies shown above horizontal gray lines. Channels are ordered by top stimulation frequency from lowest to highest. Channels without significant modulation not shown. **D** Fold-change in power relative to aperiodic fit at the peak of each identified endogenous oscillation versus the frequency of that endogenous oscillation for all identified endogenous oscillations (see Methods) across all contacts ($n = 904$ contacts, 8 subjects, 11 sessions). Dot: 1 contact's endogenous oscillation in a session. **E** Frequency of stimulation leading to maximal fold-change in power (top stimulation frequency) versus closest detected endogenous frequency, for all contacts ($n = 184$ contacts, 8 subjects, 11 sessions) that showed at least one endogenous oscillation at baseline and significant response to more than 6 of the flicker stimulation frequencies tested. Dot: one contact, dashed line: x = y, gray shaded area: +/−5 Hz from x = y. Source data are provided as a Source Data file.

manner that is specific to sensory modality (especially visual stimulation), anatomy (especially visual cortex and MTL), and general region of the SOZ (especially for TLE subjects). Flicker stimulation may thus provide focal anti-epileptic benefits with the appropriate selection of stimulation modality for the subject.

## Discussion

We studied the neurophysiological effects of sensory flicker in neurosurgery patients using invasive intracranial electrodes, a gold standard for characterizing localized neural activity. We found that flicker induces a steady-state EP across widespread brain networks, including canonical cognition-central cortices in the MTL and PFC. Moreover, we observed that flicker-induced neural responses are consistent with the resonance of long-range circuits but not linear superposition of single sensory pulse responses or the entrainment of active endogenous oscillations in recorded regions, providing mechanistic insight and a rational framework for parameter optimization. Finally, we found that multisensory flicker reduced the frequency of IEDs in epilepsy patients, especially in locations sensitive to flicker-induced changes in evoked potential and notably in the MTL of patients with temporal lobe epilepsy. The effect of flicker on IED rate depended on the modality of flicker, anatomical location of the IED, and the subjects' type of focal epilepsy (defined by locations of SOZs), implying a capacity to fine-tune flicker for specific subjects.

Flicker offers a non-invasive stimulation option which may complement other means of promoting brain oscillations, such as transcranial direct current stimulation (tDCS) and transcranial magnetic stimulation (TMS). While tDCS and TMS are increasingly applied to neuropsychological conditions with varying degrees of efficacy[92,93], TMS in particular is approved by the United States Food and Drug Administration for depression[94–96]. For maximal safety and efficacy, however, TMS requires imaging-based navigation and administration by trained clinicians, making chronic daily home use impractical. Flicker stimulation may ultimately complement these other stimulation methods by utilizing a distinct mechanism to modulate brain networks directly relevant to degenerative disorders and epilepsy. By comparison to tDCS and TMS, multi-sensory flicker features a simple, practical, and inexpensive form factor that is likely safe and effective for chronic daily home use[97]. Flicker was previously shown to modulate neural activity in rodent higher cognitive regions, reduce Alzheimer's disease-related neuropathology, and improve cognitive outcomes[69,98]. Our report likewise suggests that flicker exposure may therapeutically modulate human brain function by resonance mechanisms, laying important groundwork for future clinical development.

While many previous studies investigated steady-state EPs in humans, several were limited by either spatial or temporal resolution. Most utilized non-invasive scalp EEG[33–43], which integrates filtered and attenuated non-focal signals, with limited localization value. Others used fMRI, PET, and/or MEG, which all present various spatiotemporal

or other technical limitations[33,34,37,38,44–51]. By comparison, intracranial electrode recordings provide precise spatiotemporal resolution. Indeed, a few studies used this method to show steady-state EP in the visual system, including the lateral geniculate nucleus, the primary and secondary visual areas[52], and the broader occipital lobe[53]. Using SEEG, a standard approach to clinico-anatomico-electrophysiological correlation in which intracranial electrode arrays are placed strategically to sample widespread network dynamics, we showed that steady-state EPs occur in locally constrained neural populations across various circuits beyond canonical sensory regions. While some studies[54–58] have variably observed sensory-induced intracranial steady-state EP in insular, frontal, and temporal lobe structures, these studies were limited by low sample size, restricted testing conditions, or referencing methods which provide poor signal localization and greater potential for noise contamination. Here, we systematically tested a wide spectrum of flicker conditions (12 to 27 stimulation conditions in the different paradigms) including multiple frequencies and modalities, in a large set of subjects (19) and across widespread deep and superficial cortical regions that reflect two common categories of focal epilepsy with seizures originating in either the temporal or frontal lobes. We used Laplacian offline re-referencing of the recordings, a highly localizing method that minimizes volume conduction and noise[99]. Furthermore, we examined the responses to both periodic and aperiodic sensory stimulation and find that both have effects on neural activity with random, non-periodic stimulation modulating broad frequency ranges in some cases, while periodic stimulation induced a narrow band increase in power at the target stimulation frequency. The neural effects of random stimulation are an important consideration when designing control stimuli for flicker interventions.

We found that flicker induces steady-state EPs not only in the expected primary visual and auditory cortices, but also more broadly to involve somatomotor, limbic, and attention networks critical to supporting cognition. Previously, scalp EEG indicated strong responses to relatively low frequency (e.g., 10 Hz) visual stimulation[100]. By contrast, we detected responses at not only low (5.5 Hz), but also higher (40 Hz) frequency stimulation, which yielded widespread intracranial modulation. This observation may have resulted from key differences between scalp and intracranial EEG signals, including localization of neural activity, type of neural oscillations maximally detected, and interpretation based upon referencing and reporting metrics. Scalp EEG is an indirect measure of neurophysiological sources, particularly from the brain surface, and with limited detection of higher-frequency (gamma and above) activity due to attenuated transmission through bone and scalp. Depending in part upon the referencing method, scalp EEG represents the complex summation of filtered potentials at multiple steps along the sensory processing pathways. Consequently, scalp EEG is biased toward slower synchronized activity from the brain surface (e.g., sensory cortices), which is suboptimal for detecting steady-state EPs from small or deep neural

## A Example IEDs

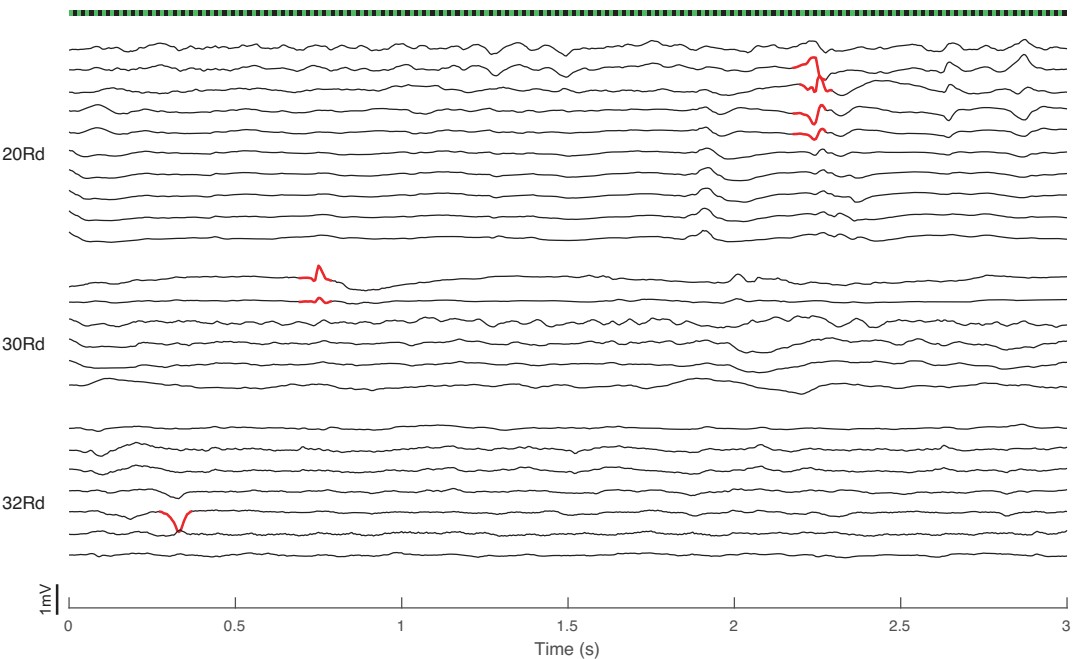

## B Effects of sensory flicker on IED rate

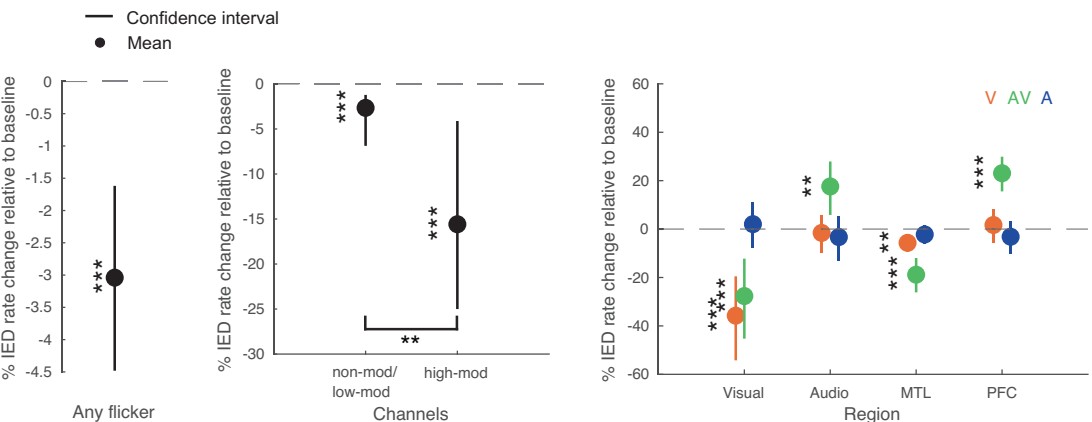

**Fig. 6 | Decrease of interictal epileptiform discharge rate in response to flicker.**
**A** Example interictal epileptiform discharges (IEDs) detected (in red) over the first 3 s of a 40Hz-audiovisual (AV) trial, across 3 depth electrodes (labeled to the left as 20 Rd, 30 Rd and 32 Rd) that had channels which detected those IEDs. Each trace represents preprocessed local field potential (LFP) signal from a contact of the depth electrode labeled to the left. Flicker stimulus shown above. **B** Left: overall effect of any sensory flicker stimulation on the IED rate when including all channels ($p = 2.4 \times 10^{-5}$; 3094 channels, 19 subjects, 25 sessions). Middle: effect of sensory flicker on IED rate in channels that were non-modulated or weakly (<1.5 fold-change in power) modulated (non-mod/low-mod, versus baseline $p = 2.8 \times 10^{-4}$; 2936 channels) or strongly (>1.5 fold-change in power) modulated channels (high-mod, versus baseline $p = 2.9 \times 10^{-4}$; 158 channels; uncorrected $p$-values are lower than Bonferroni correction for 2 comparisons) by flicker stimulation (non-mod/low-mod versus high-mod $p = 3.5 \times 10^{-3}$). Right: change in IED rate by flicker stimulation

modality (visual or V in orange, auditory or A in blue, and audiovisual or AV in green) and anatomical location of detected IEDs including visual- early visual regions, audio- early auditory regions, MTL- medial temporal lobe, PFC- prefrontal cortex (visual regions: visual flicker $p = 2.6 \times 10^{-6}$, 148 channels, 8 subjects, 9 sessions, audiovisual flicker $p = 2.1 \times 10^{-4}$, 24 channels, 2 subjects, 3 sessions; audio regions: audiovisual flicker $p = 4.5 \times 10^{-3}$, 162 channels, 7 subjects, 10 sessions; MTL: visual flicker $p = 1.8 \times 10^{-3}$, 472 channels, 17 subjects, 22 sessions, audiovisual flicker $p = 1.6 \times 10^{-8}$, 213 channels, 7 subjects, 10 sessions; PFC: audiovisual flicker $p = 3.7 \times 10^{-8}$, 261 channels, 6 subjects, 7 sessions; uncorrected $p$-values are lower than Bonferroni correction for 12 comparisons except increase in auditory regions during audiovisual flicker). For all plots, mean effect is represented by a dot, confidence interval of the effect is represented by a vertical bar; Poisson generalized linear mixed effects model for all statistical comparisons; *$p < 0.05$, **$p < 0.01$, ***$p < 0.001$. Source data are provided as a Source Data file.

populations. Also, since scalp EEG features relatively uniform signal characteristics due to distance and orientation of electrodes relative to signal sources (brain surface), such studies often report absolute change in power spectral density in response to stimulation. By comparison, intracranial EEG (e.g., SEEG) detects broad-spectrum neurophysiological activity from focal sources, irrespective of brain depth

and source-to-electrode orientation. With Laplacian re-referencing, intracranial EEG further minimizes far-field signals and volume conduction, and emphasizes focal oscillatory contribution to the steady-state EP. As intracranial EEG signals exhibit unfiltered signals from varying sources with potentially distinct neurophysiological features (e.g., white matter versus gray matter, variable geometry of electrode

placement relative to cortical laminar architecture), we presented results as the localized fold-changes in power relative to baseline, in order to normalize and compare flicker responses by conditions, brain locations, and patients. Overall, because the steady-state EP we observed may be fundamentally different from scalp EEG signals and require unique methods, future studies should contemporaneously collect and directly compare scalp and intracranial EEG responses.

Remarkably, we observed that flicker modulated neural activity in the MTL and PFC, showing that this non-invasive intervention can influence brain structures involved in higher cognitive processes and implicated in degenerative and other neuropsychiatric disorders. Although we observed significant steady-state EP in multiple cognitive regions, effect sizes were comparatively lower than in primary sensory cortices. This likely reflects the complex combination of ascending and parallel processing of sensory signals through associative and cognitive networks. Furthermore, a wider exploration of stimulation parameters, including greater intensity, could magnify steady-state EPs in higher cognitive regions such as the MTL and PFC. Even with intracranial recordings that more definitively localize recorded oscillations, the sources and routes of synchronizing input to higher cognitive regions is inferred based on anatomical connections. Visual flicker-related oscillations in MTL and PFC, for instance, are likely influenced by efferent projections from primary sensory cortices via dorsal (occipital-parietal-frontal) and ventral (occipital-temporal) visual streams[101,102], or the third visual pathway associated with the superior temporal sulcus[78]. Moreover, a strong signal recorded from contacts in the temporal lobe white matter may be due to sampling of optic radiations (Meyer's loop), positioned lateral to the human hippocampus[79].

Although we showed clear modulation of the MTL and PFC, further work is needed to determine whether such modulation might have a meaningful clinical or functional impact on cognition. There is some prior evidence that sensory flicker can impact memory consolidation[16], and current studies are pursuing potential therapeutic effects in AD. Our group's prior study showed preliminary evidence that 8 weeks of sensory flicker strengthens functional connectivity of nodes in the default mode network in patients with AD[97]. Here we found flicker modulates locations previously implicated in widespread functionally connected networks, like the somatomotor, attention, and default mode networks, that together support cognition and exhibit weakened functional connectivity in AD. Flicker stimulation may affect functional connectivity by biasing interconnected neurons to fire together[103].

To understand the pathophysiological significance of sensory flicker in patients, we assessed its effects on IEDs. IEDs are typical of epilepsy, but certain types are also observed in association with other neurological and psychiatric disorders such as AD, multiple sclerosis, autism, and ADHD[68]. In our cohort of subjects undergoing intracranial investigations of focal-onset epilepsy, we found that short (10 s) duration sensory flicker exerts a modest anti-epileptic effect across the brain (−3.0%), but with greater influence (−15.6%) in locations in which potentials were also significantly modulated by flicker. Anatomically, the strongest antiepileptic effect was by visual flicker in early visual regions (−35.8%), lending face validity to these results. Moreover, we found a significant reduction in IED rate in the MTL (−5.7% during visual flicker and −18.8% during audiovisual flicker), a region not primarily involved in sensory processing, but a principal target of therapy in epilepsy and other disorders. Importantly, IED rate was reduced in response to visual, audiovisual, and auditory flicker in the MTL of TLE subjects and in response to auditory flicker in PFC of FLE subjects, revealing potential clinically relevant reductions in IEDs. In contrast, IED rate was increased in the PFC in response to audiovisual flicker. This group-level effect may be of possibly questionable clinical impact in TLE patients, as the SOZ was not in the frontal lobe. While rhythmic photic stimulation, especially at 15–25 Hz, may elicit

seizures in certain susceptible individuals with generalized epilepsy[73], we did not observe an increase in IEDs at those visual stimulation frequencies. Importantly, our results argue against any presumption that all modalities or conditions would be generally beneficial in all persons with epilepsy. Our patient population lacked parietal lobe sampling and patients with parietal lobe epilepsy, which may be more sensitive to auditory flicker, given the role of these regions in sound localization and self-reference. We speculate that with further iteration, auditory and visual stimulation could provide anti-seizure benefits in patients with TLE. Thus, auditory and visual flicker is safe and potentially therapeutic in TLE, but further investigation is warranted, especially for patients with FLE or populations not represented in this study (e.g., focal parietal lobe, generalized epilepsies, and rare genetic epilepsies with susceptibility to photic-induced seizures[73]). Indeed, patient selection bias is an inherent limitation of our study, and flicker may have different effects in other populations. Nevertheless, our results show promise that sensory flicker conditions might be deliberately tuned to affect particular anatomical locations and types of focal seizure networks. Notably, an intervention does not need to benefit all types of epilepsy or patient populations to be of value.

Although sensory flicker might exhibit disease-modifying effects on widespread circuits associated with epilepsy, and by extension possibly AD, we note important caveats. First, we observed the impact of flicker stimulation upon IEDs to be moderate and transient, we did not optimize flicker conditions to affect IEDs, and the clinical impact of such interventions remains to be determined. We measured the effects of only brief (10 s) sensory flicker, and future studies should investigate whether longer or repeated exposures might impact meaningful clinical outcomes with respect to seizures and/or cognition. Second, since we observed correlation between the amplitude of the steady-state EP and IED reductions with relative restriction to particular circuits, multi-sensory flicker may be unsuited to influence other brain circuits. Nevertheless, optimization of flicker modulation of higher cognitive regions deserves further attention. One recent study suggested that engaging such higher cognitive regions in cognitive tasks might improve modulation extent in those regions[57]. Finally, a significant limitation of the current study is that the overall observed decrease in IED rate and focally differential effects (e.g., MTL versus PFC) under different conditions may result from general engagement of sensory circuits causing a nonspecific change in brain state, rather than a mechanism specific to sensory flicker. Future studies with additional controls are needed to fully contextualize these results.

Our study also explored the mechanisms of steady-state EP. A prior study utilizing noninvasive scalp EEG[59] suggested that the visual steady-state EP results from the linear superposition of single-pulse EPs. This proposed mechanism has been used to explain the observation that 40 Hz auditory stimulation leads to a peak in response amplitude, with the 40 Hz component of single-pulse EPs maximally summating at that frequency[39]. By contrast, our experimental paradigm using within-subject direct comparison between single-pulse EPs and steady-state EP contradicted linear superposition as a unifying mechanism. Importantly, we recorded intracranial local steady-state EPs in contrast to scalp EEG, the limitations of which were already described above. Another debated mechanism hypothesizes that the steady-state EP results from intrinsic circuit oscillatory properties, either through entrainment of active endogenous oscillations and/or resonance of stimulated circuits. In accordance with such mechanisms, we did find that specific frequencies of sensory flicker induce higher modulation in given recorded locations. These frequency preferences were not substantially related to detected active endogenous oscillations in recorded regions, which suggests that these oscillations were not entrained. This conclusion is similar to another study[63] in which MEG was used to show that visual cortex gamma oscillations (induced by viewing a grating stimulus) are not entrained by visual flicker. The authors expected an increased steady-state EP amplitude when

stimulating at a frequency close to the subject's intrinsic gamma frequency of the visual system. Instead, they observed a consistent decrease in response amplitude as a function of increasing stimulation frequency, regardless of the proximity to intrinsic gamma. Our data offers complementary findings, suggesting that sensory flicker does not seem to entrain baseline, non-stimulus induced endogenous oscillations. In contrast to the MEG study, we did find many channels across subjects showed a stronger steady-state EP for specific stimulation frequency ranges, with only a minority showing a decrease in amplitude as a function of increasing stimulation frequencies. This may be due to our directly sampling beyond visual areas, or improved signal-to-noise and spatial resolution of our recordings compared to MEG. We highlight that here, we only tested for entrainment of baseline active endogenous oscillations detected in recording sites. It is possible that sensory flicker may entrain oscillations present in nodes we did not sample (e.g., basal forebrain, thalamus, and brainstem), or that there exist more subtle, transient endogenous oscillations that we did not detect in the average power spectral density of the LFP. Moreover, a previous study[87] indicated that cases of entrainment are typically observed when the frequency of stimulation is within ~1 Hz of the endogenous oscillation frequency. Our sampling of multiple stimulation frequencies in ~3 Hz increments may be too coarse to adequately entrain sampled baseline endogenous frequencies in all cases and may contribute to absence of evidence for entrainment in our results.

Overall, we found no evidence for the linear superposition and entrainment hypotheses, but instead found evidence supporting resonance or preference of stimulated circuits to specific frequencies of stimulation. Along with examples of persistence of oscillatory activity after stimulation offset, we conclude that flicker-induced neural responses are consistent with the resonance of complex brain circuits spanning from early sensory input structures to higher level limbic and cognitive cortices. Our study shows that such putative circuits resonate to specific flicker stimulation frequencies, with many recording locations exhibiting a peak response to stimulation in specific frequency ranges. Similar to previous scalp EEG studies[100,104], we observed that the frequency preference was broadly tuned to wide frequency bands, and often with multiple peaks. This suggests that underlying circuit oscillators are broadly tuned to prefer different frequency bands. A prior rodent study[28] highlighted the existence of a local circuit oscillator. In that prior study, optogenetic stimulation of fast-spiking interneurons in the mouse sensory barrel cortex at various frequencies led to the highest increase in LFP power in the gamma range, illustrating an optimal resonant property of the local cortical circuit. Such local circuit oscillators spanning early sensory regions to our recording sites likely play a role in the observed frequency preferences. Resonance of specific oscillators may depend on the cellular composition of the circuit[84], the nature of recurrent synapses, biophysical properties of local and input neurons, and modulatory input from other circuits. Together, these results illustrate the importance of selecting an optimal frequency to maximize steady-state EP in target regions. Indeed, optimizing the amplitude of modulation in target brain regions would require using modality and stimulation frequencies preferred by the circuit that ties sensory input structures to the target region. Although we charted out which target regions respond to given stimulation frequencies, further studies will need to more carefully map the optimal resonance of stimulated neural circuits.

Despite novel findings regarding the neurophysiology of flicker and its potential therapeutic benefits in humans, there remain caveats to broad generalization. First, our subjects' brains were variably abnormal, often harboring brain lesions from previous accidents or surgeries, or various pathologies (Table S2). Patients further varied by age, sex, cognitive deficits, and baseline seizure medication regimens, although patient variation is partly mitigated by widespread brain

sampling and a relatively large group of subjects. Second, the brain regions sampled across our patients undergoing intracranial monitoring of focal epilepsy were by nature enriched for regions suspected and/or proven to harbor pathological seizure networks, raising the possibility that these observations might be specific to epileptic rather than healthy brains. This potential confound is again mitigated by widespread sampling, including brain regions that were ultimately found to be outside seizure onset zones or abnormal regions: across our 19 subjects, the majority of contacts were outside abnormal tissue and/or seizure onset zones. Third, the clinical environment in which experiments took place involved factors related to the clinical care of the subjects including varying environmental stressors, postoperative discomfort, sleep deprivation, changing medication dosages, and other factors which could affect brain states.

This study bridges findings from rodents to humans and shows that multisensory flicker non-invasively modulates brain circuits, including limbic and cognitive structures impacted in disease, potentially to therapeutic effect. This investigation is unique in using extensive direct intracranial neural recordings in humans to characterize single-pulse and steady-state EP with high spatiotemporal resolution. Furthermore, we clarified the mechanisms of steady-state EP in multiple circuits, shedding light on strategies to maximize modulation. Finally, our findings demonstrate proof of concept that flicker can reduce IEDs, a pathological brain activity associated with epilepsy, AD, autism, and other disorders. Importantly, effects of flicker upon IED rates depended on individual subjects' seizure networks, with modality of sensory flicker showing differential effects by location. This study points toward individualized tuning of non-invasive sensory stimulation to drive personalized therapy.

## Methods
### Participants
We recruited treatment-resistant epilepsy patients undergoing presurgical intracranial seizure monitoring (see Table S1 for demographics, Table S2 for epilepsy information, Table S3 for intracranial monitoring). In between clinical services and at the patient's discretion, one or more of three experimental paradigms (Figs. 1B, 4B, and 5A) were carried out in the patient's room. All study-related procedures were approved by the Emory Institutional Review Board. This interventional trial (NCT04188834) for the purpose of basic science utilized a within subject cross-over design. Each subject underwent different frequencies of flicker stimulation and periods of no stimulation in the same recording session with the order of stimulation randomized. Participants were approached by study staff before or while in the epilepsy monitoring unit and asked if they would be interested in participating in research. Recruitment criteria included: age over 18, fluent in English, able to understand and give verbal and written consent to the study procedures and associated risks, not suspected to be susceptible to photic-induced seizures or psychogenic non-epileptic seizures (PNES) triggered by sensory stimulation, did not show abnormal EEG activity if tested with clinical photic stimulation. Subjects were recruited regardless of sex (female vs male) indicated on the patient clinical chart. A total of 11 females and 8 males recruited completed or reached near completion of one or more of the paradigms in this study. Sex and gender were not considered in the design and analyses of the study, for two main reasons: (1) We did not anticipate the neurophysiological effects of sensory stimulation to differ between females and males and (2) recruited subjects are rare, due to the small number of intracranial EEG patients seen by any institution in a given time period, so we minimized parcellation of the data into smaller groups. We obtained informed consent from all recruited subjects, and 19 of the recruited subjects completed or neared completion of one or more paradigms overall (Table S4). The first and last subject were enrolled January 10, 2020 and November 21,

2022, respectively. Subjects were not compensated for participation. Based on prior studies, we anticipated that 5 to 10 subjects per group would be sufficient to test our hypotheses because our study design used within subject comparisons. No formal power analysis was performed prior to the study.

### Electrophysiology

As part of their clinical planning, patients were implanted by a neurosurgeon (JTW or REG) with SEEG depth electrodes, most often from DIXI Medical (DIXI Medical, France), with 0.8 mm diameter, 2 mm length platinum/iridium contacts, typically separated by 3.5 mm intervals center-to-center. In some subjects, depth electrodes included (usually 1–2 per candidate, consenting subject) FDA-approved Ad-Tech electrodes (Ad-Tech Instrument Corp, Racine, WI; 1.28 mm diameter, 1.57 mm length platinum contacts, separated by 5 mm intervals center-to-center) containing nine 38-micron microwires protruding from their tip that allowed recording from single neurons. The number and implant location of the stereotactic depth electrodes were exclusively determined by the clinical team and based on clinical needs. Local field potentials measured with macro-contacts were recorded on the clinical monitoring system (XLTEK EMU 128FS; Natus Medical) and associated Natus Neuroworks software, typically at a rate of 2048 Hz (range 1024–16,384 Hz). The clinical system's reference and ground were typically sub-galeal contacts from an electrode array placed subdermally at the vertex. Microwires were recorded using the Blackrock NeuroPort system (Blackrock Microsystems, UTSW) and associated Central Software Suite at a rate of 30,000 Hz using a dedicated microwire as physical reference.

### Stimulus presentation

Customized software ran in MATLAB 2019b controlled stimulus presentation, including the creation of analog sensory signals and pulses synching the EEG recordings, and control of their timing (a version of the source code is available at https://github.com/singerlabgt/Behavioral_FlickerMasterTask). These signals were produced using a digital acquisition board (USB-6212 multifunction I/O device, National Instruments), which sent analog signals to a customized circuit. Opaque glasses containing LEDs (Mind Alive Inc.) administered visual stimuli, and earbuds (SONY MDR-EX15LP) presented auditory stimuli. These glasses maximized the extent of the visual field stimulated, while earbuds maximized signal of auditory stimuli relative to surrounding noise. Visual stimuli consisted of a 50% duty cycle square wave signal, while auditory stimuli consisted of a 7 kHz tone amplitude-modulated by a pseudo-square wave envelope, with about 1.6 ms ramp up and down each to minimize noise due to amplifiers rapidly turning on and off with each cycle. We opted to use a 50% duty cycle square wave signal for sensory stimuli, as it was previously shown that such a visual square wave signal had the highest likelihood of inducing a steady-state EP in the occipital cortex, compared to other types of waves such as triangular or sinusoidal[105]. Stimulation trials were synchronized with neurophysiology acquisition systems via a TTL pulse.

At the start of any experiment, we first adjusted the brightness and volume to each subject's comfort, then administered individual trials from each sensory stimulation condition to check for any evidence of associated seizure symptoms. To control for the possibility that stimulus responses were due to an artifact from the sensory stimulation apparatus, participants underwent a relative occluded condition in which stimuli were delivered but occluded with a sleep mask and/or towels on their eyes underneath the glasses, and earplugs while earphones were near but not placed into the ears. At the end of each session, we measured the brightness and volume of the device at 40 Hz audiovisual stimulation, using a luxmeter (TRACEABLE Dual-Range Meter) and decibel meter (BAFX Products BAFX3608).

Subjects were exposed to one or more of the following three stimulation paradigms (Figs. 1B, 4B, and 5A):

**Flicker 5.5/40/80 Hz paradigm**. Subjects were exposed to 10-second trials including modalities of visual only, audio only, and audiovisual combined, at frequencies with 50% duty cycle of 5.5 Hz, 40 Hz, 80 Hz, and random pattern. Trials were pseudo-randomized (with no given modality or frequency repeated more than three times in a row) to control for order effects and minimize habituation to a given stimulus. Each stimulation trial was followed by 10 s of no stimulation, i.e., a baseline trial. In the audiovisual conditions, light and sound onset and offset were synchronized. Random pattern stimulation consisted of 12.5 ms pulses with inter-pulse intervals randomized for durations between 0–25 ms (i.e., average light exposure duration per period and average frequency was around those of the 40 Hz conditions). In total, over the about 1 h-long experiment, subjects were exposed to 360 10-second trials, with 15 trials per stimulation condition, and 180 no-stimulation (i.e., baseline) trials. During the experiment, subjects were asked to keep their eyes open, in order to maximize the visual steady-state EP, and offered breaks every 10 min. The relative occluded condition typically consisted of 6 10-second 40 Hz-visual trials and 6 10-second 40 Hz-audio trials.

**Single-pulse paradigm**. To compare flicker responses to those generated by single pulses of light or sound, subjects were exposed to 12.5 ms-long single pulses of visual only, audio only, or audiovisual modality, with inter-pulse intervals randomized between 987.5–1487.5 ms. The 12.5 ms duration of single pulses matched the duration of pulses in the 40 Hz flicker condition and thus allowed direct comparison of the responses between the two paradigms for that stimulation frequency. We here focused on the 40 Hz condition, as mouse studies focused on this condition, and scalp EEG studies have demonstrated that 40 Hz-auditory stimulation induces the highest steady-state response[39], likely due to linear superposition of single pulse evoked potentials with a 40 Hz spectral component, conveying interest in this frequency of stimulation. Each condition was repeated for a total of 200 trials. Trials from each modality were presented in a pseudorandomized manner (no given modality repeated more than three times in a row). The relative occluded condition consisted of 200 audiovisual trials.

**Flicker 5.5–80 Hz range paradigm**. To map flicker responses across a wide range of frequencies, subjects were exposed to 10-second trials of either visual or auditory modality, at frequencies spanning the 5.5–80 Hz range: 5.5 Hz, 8 Hz, 11 Hz, 14 Hz, 17 Hz, 20 Hz, 23 Hz, 26 Hz, 29 Hz, 32 Hz, 35 Hz, 38 Hz, 40 Hz, 42 Hz, 45 Hz, 48 Hz, 51 Hz, 54 Hz, 57 Hz, 63 Hz, 66 Hz, 69 Hz, 72 Hz, 75 Hz, 78 Hz, and 80 Hz. Each condition included 10 trials. We also included 10 trials of random pattern stimulation and 10 trials of no stimulation (baseline). Trials were pseudo-randomized (no given condition repeated more than three times in a row, with attempt to spread all conditions across the experiment duration) and separated by an intertrial interval randomized between 2–2.5 s. Random pattern stimulation and flicker duty cycle were the same as in the Flicker 5.5/40/80 Hz paradigm. Subjects were asked to keep their eyes open and offered breaks about every 10 min. The relative occluded condition consisted of 6 10-second 40 Hz trials of the selected modality.

### Electrode localization

As part of stereotactic planning and confirmation, subjects typically received structural T1 and T2 MRIs before the electrode implant surgery, and a CT scan and a structural T1 MRI following surgery. We identified and labeled electrodes on the post-operative CT using the voxTool software (https://github.com/pennmem/voxTool), then co-registered all imaging to pre-operative T1 MRI using rigid transformation with the Advanced Normalization Tools package (ANTs; stnava.github.io/ANTs/[106]). We calculated electrode coordinates in different imaging spaces using co-registration output and custom

MATLAB scripts that incorporated a function from Lead-DBS (lead-dbs.org[107]). Pre-operative T1 MRI was parcellated and segmented using the FreeSurfer toolbox (https://surfer.nmr.mgh.harvard.edu/[108]). Where appropriate, here and in other preprocessing steps or analyses, we used GNU Parallel to process data in parallel[109] (https://www.gnu.org/software/parallel/). Electrodes were anatomically labeled using FreeSurfer outputs and custom scripts. Anatomical label assignment was performed to identify the label from which the electrode was likely picking up the strongest signal, using a 5 mm radius Gaussian search sphere centered at the approximate center of mass of the electrode. Specifically, we (1) converted the center of each atlas voxel to RAS world coordinates, (2) searched for all anatomical labels within the search sphere around the electrode's center of mass with a radius $r$ of 5 mm, (3) removed atlas voxels labeled as cerebral white matter or white matter hypointensities, (4) for these remaining voxels, assigned a signal strength amplitude, calculated as $1/r$ (for $r = 0$, assigned value of 1), (5) summed labels and associated signal strength, (6) picked the label with the greatest overall signal strength. We extracted normalized electrode locations into MNI space, through rigid, affine then symmetric image normalization (SyN) coregistration of pre-operative T1 MRI to T1 MNI MRI (ICBM152 2009b Nonlinear Asymmetric[110,111]). Where brain imaging was used to show electrode location (Figs. 1 and 2), where appropriate we rotated the imaging volumes to the plane of the depth electrode of interest, to clearly see all contacts from the electrode.

### Identification of pathological features of recorded locations

To assess the proportion of results originating from recording locations involved in abnormal tissue or seizure network (see Discussion), we used the subjects' clinical reports, including imaging and neurophysiological reports. We identified contacts located in or near abnormal tissue (such as previous resection, encephalopathy, or periventricular nodular heterotopia), the seizure onset zone(s), and contacts implicated in the detection of interictal epileptiform discharges. When estimating the percentage of recording locations in or near abnormal tissue or seizure onset zone(s), we included any recording site which Laplacian montage included any channel tagged with such features.

### Data analysis

Most analyses were run in MATLAB 2019b, using custom scripts in combination with Fieldtrip[112] (https://www.fieldtriptoolbox.org/) and Chronux (http://chronux.org/) toolboxes.

**Exclusion of LFP channels from analysis.** We identified contacts outside brain parenchyma, in CSF, in ablation or resection lesion or cavity, and contacts that were defective or showing artifacts based on the clinical team anatomical labeling of electrodes and neurophysiology reports. In cases where anatomical labeling from the clinical team was not available, contacts outside brain parenchyma or in CSF were manually identified using pre-operative T1 MRI coregistered to postoperative CT. Moreover, for each experimental session noisy channels were identified by visual inspection of signals from a randomly selected set of 10-second segments, in both the time and frequency domains. All above channels were excluded from further preprocessing and analysis.

**Referencing of LFP recordings.** In most analyses, we aimed to localize any sensory response to the best approximation of their neurophysiological source. The monopolar referencing provided by the clinical recording system has several disadvantages, including signal and noise contamination from the subdermal physical reference placed at the vertex, and being prone to volume conduction[99]. We used Laplacian re-referencing of the LFP recordings for all analyses, as it was shown to be optimal at localizing the source of recorded signals[99,113]. The Laplacian montage is a highly localizing method, whereby volume-conducted signal across multiple contacts is minimized, while local signal is preserved. Moreover, noise resulting from movement artifact or ground noise is also greatly reduced, providing a cleaner signal. This referencing method takes the signal from each contact and subtracts the average from the two most adjacent contacts. For contacts at the extremities of depth electrodes, bipolar referencing was used (i.e., the signal from the adjacent contact was subtracted from the signal from the end contact). In cases where a channel was adjacent to an eliminated channel (due to it being noisy for instance, see above), we approximated Laplacian referencing by subtracting the mean from the two most adjacent contacts that were not eliminated. We treated bipolar referencing of end contacts with adjacent eliminated contacts similarly.

**Processing and quantification of LFP response to flicker.** For paradigms involving flicker stimulation (i.e., Flicker 5.5/40/80 Hz and Flicker 5.5–80 Hz range), LFP recordings were segmented into 12-second segments corresponding to each 10-second trial +/− one second to manage filter artifacts in a subsequent preprocessing step. For experimental sessions where there was an unequal number of trials per condition (for instance in case where the session was stopped before completion, or in the case of baseline trials for the Flicker 5.5/40/80 Hz experiment), a random subset of trials was selected for analysis for those conditions with a higher number of trials, so that all conditions had the same number of trials. Data were re-referenced as detailed above and bandpass filtered between 2–300 Hz, with a baseline correction over the duration of the 12-second segments. Power spectral density (PSD) was calculated for each 10 s trial, over 2–100 Hz, using the Chronux toolbox (http://chronux.org/), with a time-bandwidth product of 3, and number of tapers of 5.

To compare the amplitude of the steady-state EP across contacts, conditions, and subjects, for each contact and flicker condition we quantified the normalized fold-change in power at the frequency of stimulation using the following equation (similar to prior work[63]):

$$FC_{pow} = \frac{\mu_{stim}}{\mu_{bl}} - 1 \tag{1}$$

where $FC_{pow}$ is the fold-change in power at the frequency of stimulation (i.e., our measure of modulation amplitude), $\mu_{stim}$ is the power at the frequency of stimulation averaged across trials from a given condition, and $\mu_{bl}$ is the power at the frequency of stimulation averaged across an equal number of trials from the baseline condition. For each contact and condition, we also calculated a corresponding significance value of the steady-state EP, using the following method: we computed $\mu_{stim} - \mu_{bl}$, then performed a random permutation test of trial values with 10,000 iterations; cases where less than 5% of the resulting putative differences of the means were greater than the measured difference of the means, were considered significant. For illustration of the flicker response in the time domain, each 10 s trial of a given stimulation condition was re-segmented into 2 cycles of the stimulus, with one overlapping cycle between consecutive segments; in the case of the comparator, i.e., baseline condition, we likewise subdivided each 10 s trial into corresponding segments of the same length as in the stimulation condition, with also one overlapping cycle between consecutive segments. The resulting segments were averaged, and the standard error of the mean was calculated.

For statistical analysis of the proportion of contacts responding to given conditions of the Flicker 5.5/40/80 Hz paradigm (Fig. 3A top), we compared the proportions of contacts responding to visual versus audiovisual versus auditory flicker and performed a chi-square test (MATLAB crosstab function). We performed a similar analysis comparing proportions of contacts responding to 5.5 Hz versus 40 Hz versus 80 Hz (Fig. 3A middle).

For the comparison of responses to periodic and random flicker stimulation (Fig. S4), we additionally calculated in a similar fashion as above, the significance and average fold-change in power at 30 Hz, 40 Hz and 50 Hz in the random condition; we also calculated the significance and average fold-change in power at 30 Hz and 50 Hz in the 40 Hz periodic conditions. We then compared for contacts that showed significant increase in power at 40 Hz in the 40 Hz periodic conditions, the difference between fold-change at 40 Hz and average fold change difference at 30 Hz and 50 Hz:

$$FC_{diff} = FC_{pow-40Hz} - mean(FC_{pow-30Hz}, FC_{pow-50Hz}) \quad (2)$$

where $FC_{diff}$ is the mentioned difference, and $FC_{pow-40Hz}$, $FC_{pow-30Hz}$ and $FC_{pow-50Hz}$ are the fold-changes in power at 40 Hz, 30 Hz and 50 Hz, respectively. This type of calculation highlights the specificity of the response at 40 Hz (high specificity with high $FC_{diff}$, low specificity or response across broad frequency range with low $FC_{diff}$). We performed the same calculation for contacts and conditions that showed significant fold-change increase at 40 Hz in the random conditions. We then compared the distributions of $FC_{diff}$ between the periodic and random conditions.

**Processing and detection of persistence of oscillatory response.** To evaluate for persistence of oscillatory response to sensory flicker beyond the offset of stimulation, we used a method developed for this purpose[56]. Briefly, this technique applies a symmetric non-causal bandpass filter and detects the start and stop of oscillatory response to a periodic stimulus. A persistent oscillatory response is one present if there are more than one additional oscillatory cycles beyond where we would expect the oscillatory response to end based on the duration of stimulation and the delayed start of oscillatory sensory response. For example, if the start of the oscillatory response is 200 ms after the start of the stimulus, then a persistent oscillatory response would be one that continues by more than one stimulation cycle, 200 ms after the stimulus ends. Original code[56] from the study was adapted and run partly in the Python environment. Time-frequency plots represented in Fig. S8 were done using continuous wavelet transform (Matlab function cwt).

**Processing and quantification of phase-locking value.** For each contact and flicker condition, we calculated the inter-trial phase-locking value (PLV) to the stimulus, as in previous studies[63,114]. Specifically, the sensory stimulation signal was approximated with a sinusoid, then for each trial we calculated the cross-spectrogram (MATLAB's function xpectrogram) of it and the preprocessed LFP signal, with window of size half the sampling frequency, and no samples of overlap between segments. The angle difference between the two signals was then calculated for each timepoint, then averaged across trials and time, and the absolute value was used as the PLV. The significance of the angle difference across trials was assessed using Rayleigh's test for non-uniformity of circular data, calculated using a circular statistics toolbox[115]. *P*-values below 0.05 were considered significant.

**Detection of baseline endogenous oscillations.** We measured the mean frequency and amplitude of endogenous oscillations at each recording location at baseline (no stimulation). To account for the aperiodic component of the PSD, which can influence the measures of endogenous oscillations, we used the FOOOF toolbox[116] (https://github.com/fooof-tools/fooof). We used the following detection parameters on the mean baseline PSD for each recording location: max number of peaks of 5 per location, peak width limits between 2–10 Hz, minimum peak height of 0.6, and frequency range of 2–100 Hz. We then quantified the amplitude of each detected endogenous oscillation by fold-change in power of the modeled PSD at the center

frequency of a given oscillation relative to the aperiodic fit of the PSD. We compared the fold-change in power of the detected endogenous oscillation and optimal stimulation frequencies using paired-sample, two-sided *t*-tests.

**Processing of LFP single-pulse evoked response.** Recordings from the duration of the experiment +/− 60 s were re-referenced, high-pass filtered at 0.1 Hz (Butterworth IIR filter, order 4), segmented into 1 s trials +/− 0.25 s, and baseline corrected for 0.25 s before the start of trial. We then calculated the time-locked average and standard error of the LFP segments across trials. We evaluated whether a contact showed single pulse response by subtracting the root mean square of the averaged LFP from 0 (start of 12.5 ms pulse) to 1 s for the relative occluded condition from that of the stimulation condition, and performing a random permutation test of the trials with 500 iterations; cases where less than 5% of the resulting putative differences of the root mean square of the means were greater than the measured difference, were considered significant. We quantified the amplitude of the response by taking the absolute maximum peak of the response from the onset of the stimulus to 1 s after the onset of the stimulus. To compare the amplitude of the flicker response versus single-pulse response (Fig. 4D), we normalized the response to a given stimulus type (flicker or single pulse) by modality for each subject from 0.001 (minimum value) to 1 (maximum value), then took the $\log_{10}$ of those values. We assessed significant differences between the single pulse and flicker responses using paired-sample, two-sided *t*-tests.

**Anatomical characterization of the sensory response.** To describe the anatomical location of the response to sensory stimulation across brain regions and subjects, we adopted three different strategies. First, we represented the location of recording LFP contacts in 3-dimensional normalized space with associated size and color codes representing whether modulation was significant and amplitude of the response, respectively. Second, we assessed anatomical regions based on FreeSurfer-outputted anatomical labels. Finally, we plotted a heatmap of the sensory response as a function of assigned anatomical label of the electrode, and by condition and subject. For most analyses, the FreeSurfer provided Desikan-Killiany parcellation atlas and Fischl et al. 2002[117] segmentation atlas labels were used. For Fig. 1D, F, the visual group included FreeSurfer labels pericalcarine, cuneus, lingual and lateral occipital, while the auditory group included transverse temporal and superior temporal. For Fig. 2C, D, the MTL group included FreeSurfer labels temporal pole, amygdala, hippocampus, entorhinal and parahippocampal, while the PFC group included medial orbitofrontal, rostral anterior cingulate, caudal anterior cingulate, frontal pole, superior frontal, rostral middle frontal, caudal middle frontal, lateral orbitofrontal, pars orbitalis, pars triangularis and pars opercularis. For analyses involving functional networks, we used surface-to-surface coregistration of Yeo et al. 2011's labels[81] provided by FreeSurfer in their fsaverage space, to individual subjects. These labels correspond to a set of 7 networks clustered via resting state functional connectivity across 1000 healthy subjects. Here, and elsewhere, final figure panels were outputted in part using code from the export_fig toolbox (https://github.com/altmany/export_fig). Moreover, violin plots were produced using code from the Violinplot-Matlab toolbox[118] (https://github.com/bastibe/Violinplot-Matlab).

**Processing of neuronal unit response to sensory stimulation.** For each microwire recording, spikes were extracted and clustered using the Combinato Python-based software[119] (https://github.com/jniediek/combinato), with threshold for extraction six times the standard deviation of noise. We then manually classified outputted groups of sub-clusters as artifact, putative multi-unit, or putative single neuron using criteria similar to previously defined[120], detailed in Table S5. A group was classified as a single unit if it satisfied all criteria, as a multi-

unit if it did not meet any of the artifact criteria, and as an artifact otherwise. Moreover, a group was considered an artifact if events tended to occur within confined periods of the experiment.

For analysis of the effects of flicker on spiking activity, ten second trial segments were re-segmented into two stimulus cycles, with one cycle overlapping between pairs of cycles. In the case of random flicker condition, cycles were composed of adjacent 25 ms segments (again, with one overlapping segment between pairs of cycles). For each unit, we then calculated the peristimulus-time histogram aligned to the onset of each pair of stimulus pulses. Units with no spikes for more than 20% of the peristimulus time histogram bins in all conditions were deemed to have too few spikes to assess whether they were modulated by sensory flicker, and thus were eliminated from further analyses. To determine the strength of spiking modulation by stimulus phase of each unit, we calculated the vector strengths and Rayleigh statistics of units for each stimulus condition using a circular statistics toolbox[115].

**Modeling of the linear superposition of single pulse evoked potentials.** We modeled the linear superposition of single pulse evoked responses by generating simulated flicker responses from summed single pulse responses. We generated 15 10 s 40 Hz simulated flicker trials (visual, audiovisual, and auditory) by linearly summing randomly selected (among 200 trials) single-pulse EPs every 25 ms. This is analogous to the 40 Hz flicker stimulation, with 12.5 ms pulses repeated every 25 ms. Using randomly selected trials (as opposed to the averaged single pulse response) accounted for variability of the response to single pulses. The amplitude of the simulated flicker response was then calculated in the same way as for the recorded flicker response (Fig. 4E). We assessed significant differences between the real and simulated data using paired-sample, two-sided *t*-tests.

**IED detection and analysis.** Manual detection of IEDs is a time-consuming process requiring expert clinical input. We thus opted to use a previously validated automated IED detection algorithm[71,91,121]. Channels eliminated for the flicker modulation analysis (because of being outside brain parenchyma, noisy, or other reasons mentioned above) were also eliminated before performing preprocessing and IED detection. Spikes occurring within a time window of 100 ms were considered part of the same IED (i.e., same spike detected by close-by channels, or traveling spike detected by distant channels, or rapidly occurring train of spikes), and IEDs detected across more than 11 channels were deemed to be noise and eliminated, as was done in a previous study[71].

We wanted to assess the effects of sensory flicker on IED rate for both the Flicker 5.5/40/80 Hz and Flicker 5.5–80 Hz range tasks. To assess for the effects of various flicker conditions on IED rate and by groups of channels (i.e., by anatomical region where IEDs occurred or groups of channels based on flicker-response), we opted to use a Poisson generalized linear mixed effects model, as used in a previous study running similar analyses[122]. This statistical model is ideal for our outcome count variable (i.e., IED count per 10 s trial) and to account for several confounding variables, such as variable IED rates by brain location, across given sessions, between sessions and between patients. Specifically, we identified independent variables of interest: flicker condition (versus baseline), anatomical location of detected IEDs, flicker modulation amplitude of where IEDs were detected, and independent confounding variables subject, task (Flicker 5.5/40/80 Hz and Flicker 5.5–80 Hz range), session number, and trial start time. The frequency of detected IEDs was generally highly variable on the timescale of minutes. To control for this variability, we included as a confounding variable the start time of each stimulation trial and assigned the same time value for the adjacent baseline segment (i.e., matching in IED rate for that time in the experiment). For the Flicker 5.5/40/

80 Hz task, the baseline segment corresponded to the 10 s baseline trial following a given stimulation trial. In the case of the Flicker 5.5–80 Hz range task, because there were only a few 10 s baseline trials across the experiment, the matching baseline segment corresponded to the closest 5 2 s baseline stimulation-OFF periods that were interspersed among flicker condition trials. For individual tests examining the effect of a flicker condition on IEDs vs. baseline (i.e., modality, frequency, or modulation level), we stratified the data by selecting the corresponding samples and performing a separate test, rather than fitting one model for all conditions. To analyze the effects of flicker on IEDs by brain region or group of channels, in cases where spikes were detected by multiple channels, they were assigned to the first channel detecting the first spike in that IED event. For the analysis segregating results by brain region (early visual, early auditory, MTL and PFC), the same FreeSurfer anatomical labels were used to select channel groups (as in Figs. 1 and 2; see Methods above for more details), and IED counts for each individual trial were summed only across those channels from each group. For the analysis segregating channels by degree of modulation, we separated them into those that were not significantly modulated or had low modulation, below 1.5 fold-change in power (non-mod/low-mod in Fig. 6), and those that were significantly modulated, above 1.5 fold-change (high-mod). The statistical test remained significant with similar results when picking from a range of nearby threshold values (between 0.5 and 2). For all results, the estimated mean and confidence intervals for IED percent change were computed from the estimated coefficients of the Poisson generalized linear model equation: $\mu_\% = 100*(e^\mu - 1)$. These values were represented in Figs. 6B, and S9B–D. We used this approach and visualization because IED data are discrete-valued, low-frequency, and Poisson-distributed, so estimates of the relative change versus baseline are often ill-defined and unstable (e.g., a change in IED count of 1 to 2 events would be 100% increase; a change of 0 to 1 would be an infinite increase). This effect is exacerbated by the spatial sparsity of SEEG data: many patients may have 0 or near-0 IED count trials when they are counted for a given brain region, or when a patient has very few channels in a given region. Computing and plotting per-session means of the IED percent change therefore produces misleading statistical outliers, making this visualization a poor representation of statistical effects in the data. To provide a more accurate representation of the data, we explicitly model both the discrete-valued nature of IED values as well as patient heterogeneity using Poisson generalized linear mixed-effects models and quantify statistical differences. Figures 6A, S9B–D demonstrate IED percent change using the coefficients and confidence intervals of these Poisson generalized linear mixed-effects models. To give an overall appreciation for the changes in IED rate by experimental session in response to any flicker, excluding the contribution of other variables, we have also represented the average percent changes between mean baseline IED count and mean flicker IED count, per session, in Fig. S9A. To run analyses by subgroup of patients based on their SOZ location, i.e., temporal lobe epilepsy (TLE) versus frontal lobe epilepsy (FLE), we classified subjects by respective group (Table S2) based on the final neurologist note on the determined SOZ.

## Statistical approach
Details on the statistical approach are described with each analysis. To determine significant differences between stimulation conditions, most analyses utilized paired-sample, two-sided *t*-tests. To compare the proportion of contacts modulated during different stimulation conditions, we used chi-square statistic of the difference between proportions.

## Reporting summary
Further information on research design is available in the Nature Portfolio Reporting Summary linked to this article.

## Data availability

Source data for all figures are provided with this paper. Minimally processed neurophysiological data, stimulation information, and electrode locations generated in this study have been deposited in the Data Archive for the Brain Initiative (DABI, https://dabi.loni.usc.edu) under project code BM2ZIVWKBFH8 and identifier https://doi.org/10.18120/4bfr-1x58. Brain imaging data are protected and are not available due to data privacy concerns. Individual de-identified data is shared in supplementary tables (sex, language dominance, anti-epileptic medication, preoperative imaging findings, determined seizure focus, IED rate, and seizure events) and in the DABI database (neurophysiological data, stimulation exposure, and electrode placement). Age is provided in aggregate in the supplement to protect subjects' privacy. The study protocol is available on clinicaltrials.gov (NCT04188834). Data from this study is available for research purposes. Source data are provided with this paper.

## Code availability

Code used for analysis of the minimally preprocessed data is available on GitHub at https://github.com/singerlabgt/MultisensoryFlickerHumanIntracranial. A version of the code used for generating stimulation paradigms is available on GitHub at https://github.com/singerlabgt/Behavioral_FlickerMasterTask.

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

## Acknowledgements

We are grateful to the patients and clinical staff of the Emory University Hospital Epilepsy Monitoring Unit for their participation and assistance with the study. A.C.S. acknowledges the Packard Foundation, NIH NINDS R01 NS109226, NIH NINDS RF1NS109226, NIH NIA RF1AG078736, the McCamish Foundation, and the Lane Family; J.T.W. acknowledges NIH NIMH R01 MH120194 and NIBIB P41 EB018783. L.T.B. was in part supported by DARPA-BAA-14-08, U01NS113198, NIMH R01 MH120194 and the Georgia Tech/Emory NIH/NIBIB Training Program in Computational Neural-engineering (T32EB025816); E.R.C. was in part supported by the Emory Neuromodulation Technology Innovation Center (ENTICe) and a McCamish Parkinson's Disease Innovation Program Blue Sky Grant. The funding sources had no role in the design, analysis, and interpretation of the study nor in the manuscript writing or decision to publish. We thank members of the Singer laboratory and National Center for Adaptive Neurotechnologies (NCAN) at Washington University, as well as Dr. Leonardo Bonilha for feedback on the methods, results, and manuscript.

## Author contributions

L.T.B., J.T.W., and A.C.S. designed the study; L.T.B. performed research and analyzed the data; E.R.C. performed most of the analysis on the effects of flicker on IED rate; E.C. and J.K.P. assisted in localizing electrode contacts; M.Y.W. built the circuit for the sensory stimulation device; J.T.W. and R.E.G. implanted electrodes in subjects; B.T.C consulted on IRB protocol and recruitment of candidate subjects, especially in terms of their risk for photic-induced seizures; L.T.B., J.T.W., and A.C.S. wrote the paper.

## Competing interests

A.C.S. owns shares of and serves on the Scientific Advisory Board of Cognito Therapeutics. A.C.S. is an inventor on allowed U.S. Patent Application No. 16/979,226. Her conflicts are managed by Georgia Tech. The remaining authors declare no competing interests.
