## [Peer Review File · Nature Communications]

Multisensory flicker modulates widespread brain networks and reduces interictal epileptiform dischargesEditorial Note: This manuscript has been previously reviewed at another journal that is not operating a transparent peer review scheme. This document only contains reviewer comments and rebuttal letters for versions considered at *Nature Communications*. Mentions of the other journal have been redacted.

REVIEWER COMMENTS

Reviewer #2 (Remarks to the Author):

This work was reviewed previously for [REDACTED] I agree Nature Communications is a more appropriate journal. This reviewer's prior concerns have been addressed in this revised manuscript.

Reviewer #3 (Remarks to the Author):

I would like to thank the authors to address each of my comments in their revision. I think they have done a very good job in addressing all of my comments. I have only one minor comment left that the authors may want to address. Otherwise, I think this is an excellent manuscript, which I would be delighted to see published.

Minor point:

The authors write in their results section:

"In particular a previous study⁸ suggested that modulating brain oscillations at 5.5Hz or theta-like frequency may improve memory consolidation, while fast- gamma is also thought to be involved in memory. Random stimuli served as a non-periodic control."

This statement is lacking a description of how fast gamma is supposed to be involved in memory. The authors could cite works by Laura Colgin showing that fast gamma is associated with memory encoding processes to better motivate their choice.

Reviewer #4 (Remarks to the Author):

The authors have improved their manuscript, and I believe this will be a useful contribution addressing basic science questions about the neurophysiological mechanism of sensory flicker. However, the clinical / therapeutic value of this approach, especially relative to other neuromodulation approaches, remain a major weakness and should be corrected.

1. My concerns regarding inaccurate statements about other competing neuromodulation technologies are not adequately addressed – and in fact are now made worse in this revision. For example, the statement in the abstract that “many non-invasive interventions have limited impact on deep cortical structures or are impractical for chronic use” is not valid. The evidence that TMS has impact on deep cortical structures (likely mediated by connectivity to the stimulation site) is just as good if not better than the current evidence re sensory flicker. Similarly, tDCS has multiple papers supporting the practicality and tolerance of chronic daily use, while sensory flicker has none. The statements in the discussion that TMS and tDCS are “limited in their therapeutic application”, require “imaging-based navigation”, have “safety concerns”, with “effectiveness that is poorly defined” are all incorrect. TMS has multiple FDA approved indications, a proven safety record, does not require neuronavigation, and is supported by multiple double-blind randomized controlled trials showing efficacy.

2. I do not follow their rationale for not correcting for multiple comparisons. I suggest dedicated stats review if proceeding with publication.

3. In response to my concern that they used different controls for different analyses, the authors chose to use baseline as a control for all analyses and dropped the random stimulation, presumably because using random stimulation as a control didn't produce significant results for some of their experiments. However, the latter is a much better control for non-specific effects of stimulation. If they are unable to use random stimulation as a control (perhaps because it was not performed for all experiments) this should be highlighted as a major limitation.

We thank the reviewers for the valuable comments and enthusiasm for the paper. We have further revised the manuscript to provide a more balanced view of the strengths and weaknesses of other non-invasive stimulation techniques, to clarify the rationale for the comparisons used and to highlight limitations of our own study. We address these additional concerns in detail below in blue text.

REVIEWER COMMENTS

Reviewer #2 (Remarks to the Author):

This work was reviewed previously for [REDACTED]. I agree Nature Communications is a more appropriate journal. This reviewer's prior concerns have been addressed in this revised manuscript.

We thank the reviewer for their prior insightful comments.

Reviewer #3 (Remarks to the Author):

I would like to thank the authors to address each of my comments in their revision. I think they have done a very good job in addressing all of my comments. I have only one minor comment left that the authors may want to address. Otherwise, I think this is an excellent manuscript, which I would be delighted to see published.

Minor point:

The authors write in their results section:

"In particular a previous study⁸ suggested that modulating brain oscillations at 5.5Hz or theta-like frequency may improve memory consolidation, while fast- gamma is also thought to be involved in memory. Random stimuli served as a non-periodic control."

This statement is lacking a description of how fast gamma is supposed to be involved in memory. The authors could cite works by Laura Colgin showing that fast gamma is associated with memory encoding processes to better motivate their choice.

The reviewer makes a good point. Indeed, work by Dr. Colgin has been inspirational to our studies. We have revised the manuscript accordingly:

"...while fast- gamma is also thought to be involved in memory (Colgin *et al Nature* 2009, Bieri *et al Neuron* 2014, Colgin *Current Opinion in Neurobiology* 2015)"

Reviewer #4 (Remarks to the Author):

The authors have improved their manuscript, and I believe this will be a useful contribution addressing basic science questions about the neurophysiological mechanism of sensory flicker. However, the clinical / therapeutic value of this approach, especially relative to other neuromodulation approaches, remain a major weakness and should be corrected.

1. My concerns regarding inaccurate statements about other competing neuromodulation technologies are not adequately addressed – and in fact are now made worse in this revision. For example, the statement in the abstract that “many non-invasive interventions have limited impact on deep cortical structures or are impractical for chronic use” is not valid. The evidence that TMS

has impact on deep cortical structures (likely mediated by connectivity to the stimulation site) is just as good if not better than the current evidence re sensory flicker. Similarly, tDCS has multiple papers supporting the practicality and tolerance of chronic daily use, while sensory flicker has none. The statements in the discussion that TMS and tDCS are “limited in their therapeutic application”, require “imaging-based navigation”, have “safety concerns”, with “effectiveness that is poorly defined” are all incorrect. TMS has multiple FDA approved indications, a proven safety record, does not require neuronavigation, and is supported by multiple double-blind randomized controlled trials showing efficacy.

The reviewer makes an important point. We have revised the manuscript to address these points. In particular, we highlight that TMS and tDCS have different strengths and weaknesses and flicker stimulation is one added method to complement these. We think that part of the issue with the prior version of the manuscript is that we tried to summarize the limitations of TMS and tDCS too briefly. In the abstract we instead focus on the need for technologies that are BOTH accessible for chronic home use and target deep structures. We have also added a citation describing the feasibility of chronic daily flicker stimulation (He et al 2021).

In the abstract:

“Modulating brain oscillations has strong therapeutic potential. Interventions that both non-invasively modulate deep brain structures and are practical for chronic daily home use are desirable for a variety of therapeutic applications.”

In the discussion:

“Flicker offers a non-invasive stimulation option which may complement other means of promoting brain oscillations, such as transcranial direct current stimulation (tDCS) and transcranial magnetic stimulation (TMS). While tDCS and TMS are increasingly applied to neuropsychological conditions with varying degrees of efficacy^{93,94}, TMS in particular is FDA approved for depression^{95–97}. For maximal safety and efficacy, however, TMS requires imaging-based navigation and administration by trained clinicians, making chronic daily home use impractical. Flicker stimulation may ultimately complement these other stimulation methods by utilizing a distinct mechanism to modulate brain networks directly relevant to degenerative disorders and epilepsy. By comparison to tDCS and TMS, multi-sensory flicker features a simple, practical, and inexpensive form factor that is likely safe and effective for chronic daily home use⁹⁸.”

2. I do not follow their rationale for not correcting for multiple comparisons. I suggest dedicated stats review if proceeding with publication.

We agree with the reviewer’s desire to ensure the statistical approach used is rigorous and accurate. Considering this reviewer’s point, we have re-evaluated our statistical approach and rationale. Correcting for multiple comparison is used in situations where one is testing if there are differences between multiple groups means or medians (or similar aspect of a distribution). This approach would be appropriate if we were claiming that the fold-change in modulation during 5.5Hz stimulation is significantly higher than 40Hz and 80Hz, like in a particular brain region or contact. We do not make this claim and rather focus most of our results on 40Hz, as the reviewer noted earlier. When we do compare different stimulation types, we report differences in the proportion of contacts that respond to different modalities and frequencies of flicker (Fig. 3A). We erred previously in that we reported our observations without statistical tests to support them. We apologize for this oversight. We have now added statistical tests, specifically chi-square test for homogeneity of proportions, that show indeed there are significant differences in the proportion of modulated channels in response to audio-visual, visual, and audio flicker and to 40Hz, 5.5Hz, and 80Hz flicker.

Revised results now state:

“Overall, audio-visual flicker produced the broadest responses, i.e., more contacts with significant steady-state evoked potential, across all frequencies tested, followed by visual flicker and auditory flicker (chi square statistic of the difference between proportions of modulated contacts 46.9, p -value= 6.6×10^{-11} , $df=2, 3$ proportions, 2067 channels included; Figure 3A, top). With respect to frequencies of stimulation tested, more contacts exhibited a steady-state EP in response to 40Hz, than to 5.5Hz or 80Hz stimulation (chi square statistic of the difference between proportions of modulated contacts 47.8, p -value= 4.1×10^{-11} , $df=2, 3$ proportions, 2067 channels included; Figure 3A, center).”

Later in the manuscript we investigate if contacts have higher modulation at one or a subset of frequencies (Figure 5). We characterize the responses to each stimulation frequency and visualize these effects, but we do not make statistical claims that modulation in response to one frequency is significantly higher than another.

3. In response to my concern that they used different controls for different analyses, the authors chose to use baseline as a control for all analyses and dropped the random stimulation, presumably because using random stimulation as a control didn't produce significant results for some of their experiments. However, the latter is a much better control for non-specific effects of stimulation. If they are unable to use random stimulation as a control (perhaps because it was not performed for all experiments) this should be highlighted as a major limitation.

We would first like to assure the reviewer that we did not select our control because “random stimulation didn't produce significant results.” We decided on no stimulation as the primary control for several reasons and we include a detailed analysis of random stimulation to help the field understand the difference between these types of stimuli. We use no stimulation as our main control because our primary question is how does flicker stimulation compare to no stimulation. Random itself is not a neutral or no stimulation condition. Indeed, we show that random stimulation has its own effects. In the prior revision we added a more detailed characterization of random stimulation to help readers understand the differences between periodic and non-periodic flicker stimulation. Random stimulation is an appropriate control when the primary question is about whether the effect of a periodic stimulus differs from a non-periodic stimulus. Specifically, our main question was if and where flicker increases power in the frequency of flickering stimulus in humans. Both periodic and random flicker has frequency components (e.g., turn on and off at some frequencies), the difference is that periodic flicker has a very narrow frequency band while random flicker has a wide frequency band. Thus, we would expect both would increase power in the frequency of the flickering stimulus, for a narrow band for periodic flicker and for a wide band for random flicker. Indeed, that is what we show (Fig. S4). We now lay out this rationale more clearly in the results and clearly state that we report both periodic and aperiodic flicker have effects on neural activity in the discussion.

In the Results:

“To contrast the responses to periodic versus random flicker stimulation, we compared the specificity of modulation at the frequency of stimulation of 40Hz versus random conditions. Both periodic and random flicker have frequency components (the frequencies at which the stimuli turn on and off), the difference being that periodic flicker has a very narrow frequency band while random flicker has a wide frequency band. Thus, we would expect that both conditions would increase power in the frequency of the flickering stimulus: a narrow band for periodic flicker and a wide band for random flicker. Indeed, random, non-

periodic stimulation induced increases in LFP power in broad frequency ranges when a strong sensory response was present, while periodic stimulation induced a narrow band increase in power at the frequency of stimulation (Figure S4).”

In the Discussion:

“Furthermore, we examined the responses to both periodic and aperiodic sensory stimulation and find that both have effects on neural activity with random, non-periodic stimulation modulating broad frequency ranges in some cases, while periodic stimulation induced a narrow band increase in power at the target stimulation frequency. The neural effects of random stimulation are an important consideration when designing control stimuli for flicker interventions.”

For the later IED analysis we noticed that IEDs decreased relative to baseline across a wide range of flicker frequencies including random stimulation, thus we did not have a strong rationale to analyze these types of flicker separately. We now note this is a significant limitation in the Discussion:

“Finally, a significant limitation of the current study is that the overall observed decrease in IED rate and focally differential effects (e.g. MTL versus PFC) under different conditions may result from general engagement of sensory circuits causing a nonspecific change in brain state, rather than a mechanism specific to sensory flicker. Future studies with additional controls are needed to fully contextualize these results.”